# Distinct homeostatic modulations stabilize reduced postsynaptic receptivity in response to presynaptic DLK signaling

Pragya Goel [1,2] & Dion Dickman [1]

Synapses are constructed with the stability to last a lifetime, yet sufficiently flexible to adapt during injury. Although fundamental pathways that mediate intrinsic responses to neuronal injury have been defined, less is known about how synaptic partners adapt. We have investigated responses in the postsynaptic cell to presynaptic activation of the injury-related Dual Leucine Zipper Kinase pathway at the *Drosophila* neuromuscular junction. We find that the postsynaptic compartment reduces neurotransmitter receptor levels, thus depressing synaptic strength. Interestingly, this diminished state is stabilized through distinct modulations to two postsynaptic homeostatic signaling systems. First, a retrograde response normally triggered by reduced receptor levels is silenced, preventing a compensatory enhancement in presynaptic neurotransmitter release. However, when global presynaptic release is attenuated, a postsynaptic receptor scaling mechanism persists to adaptively stabilize this diminished neurotransmission state. Thus, the homeostatic set point of synaptic strength is recalibrated to a reduced state as synapses acclimate to injury.

[1] Department of Neurobiology, University of Southern California, Los Angeles, CA 90089, USA. [2] USC Graduate Program in Molecular and Computational Biology, University of Southern California, Los Angeles, CA 90089, USA. Correspondence and requests for materials should be addressed to D.D. (email: dickman@usc.edu)

Neurons are endowed with robust surveillance systems that detect injury and initiate latent plasticity programs involving regenerative and degenerative responses. A fundamental signaling system induced after neuronal injury is mediated by an evolutionarily conserved mitogen-activated protein kinase called Dual Leucine Zipper Kinase (DLK). DLK signaling initiates translational changes in axons and transcriptional responses in the nucleus that ultimately promotes degeneration at the distal axon and regeneration proximal to the site of injury[1–5]. Members of the Phr1/Highwire/Rpm-1 (PHR) protein family control DLK signaling to gate neuronal injury-related signaling programs[3,5–9]. At the *Drosophila* neuromuscular junction, *highwire* (*hiw*) encodes an E3 ubiquitin ligase that constitutively degrades the DLK homolog Wallenda (Wnd)[10,11]. However, after axonal injury, Wnd is no longer degraded by Hiw, leading to increased Wnd protein levels and activation of regenerative and degenerative signaling programs[9,12]. Genetic loss of *hiw* in neurons constitutively activates Wnd signaling, while neuronal overexpression of *wnd* can overcome Hiw-mediated degradation to activate this same program[9,10,12]. This relationship between PHR proteins, DLK activity, and injury-related signaling is conserved in other invertebrate and mammalian systems[3,6,7,13–16]. Thus, loss of *hiw* or overexpression of *wnd* in neurons activates an intrinsic signaling system that transforms the cell into a state of persistent degenerative and regenerative adaptations related to a programmed response to injury.

Although, Wnd/DLK signaling and other injury-related responses occur cell autonomously and are intrinsic to the specific neuron, there is emerging evidence that synaptically connected cells may sense this perturbation and adapt in response. For example, foundational studies at the mouse NMJ have demonstrated that motor neuron injury, denervation, or synapse elimination can provoke disassembly and remodeling of the postsynaptic specialization, including a diminution of neurotransmitter receptor levels, which can precede obvious changes in the overlying nerve terminal[17–19]. There is also evidence for neuronal and synaptic remodeling in the spinal cord and other areas in the central nervous system following injury[20–23]. At the glutamatergic *Drosophila* NMJ, an apparent reduction in the synaptic response to glutamate (quantal size) was shown in *hiw* mutants as well as following a "nerve crush" injury to otherwise wild-type NMJs[9,24]. Neuronal expression of *hiw* restores normal synaptic strength and quantal size in *hiw* mutants[25], indicating that while the postsynaptic muscle does not itself experience *hiw*-related signaling, a reduction in either the amount of glutamate released and/or the postsynaptic sensitivity to neurotransmitter occurs in response. However, it remains unclear how synaptic function and plasticity change in response to presynaptic Wnd/DLK signaling, a state in which injury-related signaling is active but before complete degeneration or loss of the presynaptic terminal has occurred.

At the *Drosophila* NMJ, two forms of homeostatic plasticity have been described that stabilize synaptic strength in response to perturbations that would otherwise disrupt functionality. First, pharmacological or genetic manipulations that diminish postsynaptic glutamate receptor (GluR) function initiate a retrograde signaling system. Specifically, a retrograde signal emitted from the muscle instructs the neuron to compensate by increasing presynaptic glutamate release to restore set point levels of synaptic strength[26,27]. This process is conserved in rodents and humans[28,29], and is termed presynaptic homeostatic potentiation (PHP) because the expression mechanism of this form of plasticity is a presynaptic increase in neurotransmitter release. Second, hypo-innervation of the NMJ induces a distinct form of adaptive plasticity. In this situation, a reduction in presynaptic neurotransmitter release is observed proportional to the reduction in

synapse number[30]. In response, a homeostatic increase in quantal size is induced that stabilizes synaptic strength[30]. However, to what extent these homeostatic mechanisms operate at synapses that have adapted to injury-related signaling is unknown.

We have characterized synaptic structure, function, and plasticity at NMJs with active Wnd signaling in presynaptic neurons, with a particular focus on how the postsynaptic muscle responds. This analysis has revealed that the postsynaptic muscle responds by diminishing GluR abundance and by silencing the retrograde homeostatic signaling system that would normally enhance presynaptic release following reduced GluR levels. However, this subdued state of synaptic strength is homeostatically maintained through adaptive modulations to postsynaptic GluR levels following hypo-innervation. Together, this illuminates the distinct signaling systems targeted for modulation in the postsynaptic cell that stabilize a muted synaptic state in response to presynaptic Wnd signaling.

## Results

**Presynaptic Wnd signaling reduces postsynaptic GluR levels.** We first sought to address what specific impacts neuronal Wnd signaling has on pre- and post-synaptic function. To accomplish this, we examined four genetic conditions (Fig. 1a): wild type (*w1118*) serves as our control genotype to compare to null mutations in *hiw* (*hiwΔN*), a condition of constitutively active Wnd signaling[10]. However, *hiw* mutants have been shown to have Wnd-independent functions[10,31]. We therefore utilized two additional conditions to define Wnd signaling specifically in neurons, separate from Wnd-independent roles of Hiw in neurons and other cell types. First, we overexpressed *wnd* in neurons (wnd-OE) to activate Wnd signaling without genetic loss of *hiw*. Second, we examined *hiw;wnd* double mutants, a condition that terminates all Wnd-related neuronal signaling but retains any potential Wnd-independent functions of Hiw. Both *hiw* mutants and wnd-OE exhibit the exuberant synaptic growth and reduced intensity of presynaptic components characteristic of active Wnd signaling, with enhanced neuronal membrane surface area, reduced vesicular glutamate transporter (vGlut) intensity, reduced active zone density, and slightly reduced active zone/GluR apposition (Supplementary Fig. 1a–e).

To define the functional changes at synapses induced through neuronal Wnd signaling, we characterized NMJ electrophysiology. *hiw* mutants exhibit a reduction in both mEPSP and EPSP amplitude (Fig. 1a, b), as previously shown[10,11]. We observed a similar reduction in mEPSP and EPSP amplitude in wnd-OE compared to wild type, as well as to a driver only control (*c380-Gal4/+*; Fig. 1a, b; Supplementary Table 1). However, while the reduction in mEPSP amplitude observed in *hiw* was suppressed in *hiw;wnd* double mutants, EPSP amplitudes remained depressed, as previously reported[10] (Fig. 1a, c), with a concomitant reduction in quantal content (Fig. 1d). This demonstrates that Hiw has a previously described Wnd-independent role in promoting presynaptic neurotransmitter release, but does not directly affect mEPSP amplitude. In contrast, Wnd signaling alone in neurons reduces mEPSP amplitude but has no significant impact on presynaptic neurotransmitter release (quantal content).

Presynaptic Wnd signaling (*hiw* mutants or wnd-OE) could, in principle, decrease the amount of glutamate released from each individual synaptic vesicle (a presynaptic change) or down-regulate postsynaptic GluR levels and/or function to reduce mEPSP amplitude. We therefore investigated the state of postsynaptic GluRs. At the fly NMJ, two receptor subtypes containing GluRIIA- or GluRIIB subunits form complexes with the common GluRIIC, GluRIID, and GluRIIE subunits and mediate the postsynaptic currents driving neurotransmission[32,33].

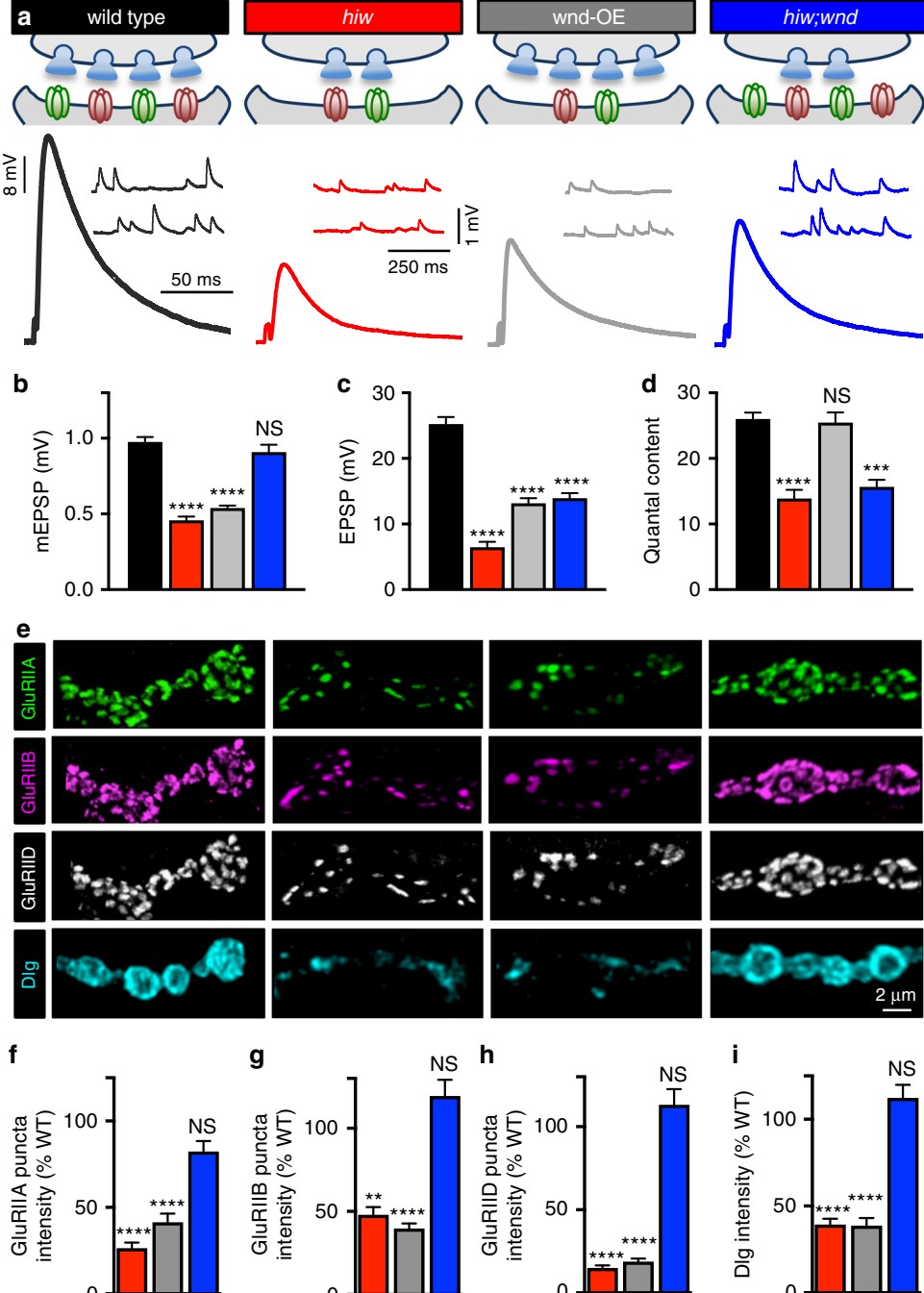

**Fig. 1** Postsynaptic reduction in sensitivity to neurotransmitter and GluR remodeling in response to presynaptic Wnd signaling. **a** Schematic of genetic manipulations to the *Drosophila* NMJ that activate or silence presynaptic Wnd signaling. Wild-type synapses (*w[1118]*) serve as the control condition. *highwire* mutants (*hiw*: *hiw[ΔN]*) show reductions in postsynaptic sensitivity and presynaptic neurotransmitter release. Neuronal overexpression of Wnd (wnd-OE: *c380-Gal4/+; UAS-wnd/+*) results in a similar reduction in postsynaptic sensitivity but normal presynaptic neurotransmitter release (compared to wild type and driver only control: *c380-Gal4/+*; see Supplementary Table 1). Switching off Wnd signaling in *hiw* mutants (*hiw;wnd: hiw[ΔN];wnd3/wndDf*) suppresses the reduced postsynaptic sensitivity but fails to suppress the reduced presynaptic release. Inset: Representative mEPSP and EPSP electrophysiological traces of third-instar larval NMJs in the indicated genotypes. Quantification of mEPSP amplitude (**b**), EPSP amplitude (**c**), and quantal content (**d**) in the indicated genotypes. Note that while the reduced mEPSP amplitude observed in wnd-OE and *hiw* mutants is suppressed in *hiw;wnd* double mutants, EPSP amplitude and quantal content fails to be restored in this genotype, suggesting a Wnd-independent role for Hiw in regulating presynaptic release but a Wnd-dependent role in modulating postsynaptic sensitivity to neurotransmitter. **e** Representative images of muscle 4 NMJs immunostained with antibodies against three postsynaptic glutamate receptor subunits (GluRIIA: green; GluRIIB: magenta; GluRIID: white) or Disc-Large (Dlg: blue). Quantification of individual GluRIIA (**f**), GluRIIB (**g**), and GluRIID (**h**) puncta sum intensity (total fluorescence of each individual GluR puncta) and **i** average intensity of Dlg staining encapsulating each bouton normalized to wild-type reveals reduced levels of this postsynaptic scaffold and receptors induced by presynaptic Wnd signaling in *hiw* mutants and wnd-OE. This response is suppressed in *hiw;wnd* double mutants. Error bars indicate ± SEM. One-way analysis of variance (ANOVA) test was performed, followed by a Tukey's multiple-comparison test. **p 0.01; ***p = 0.001; ****p = 0.0001, NS not significant, p > 0.05

We quantified the total GluR puncta number and fluorescence intensity of both GluRIIA and GluRIIB subunits, as well as the common glutamate receptor subunit GluRIID in *hiw*, wnd-OE, and *hiw;wnd* double mutants. We observed a significant reduction in the mean intensity and area of each individual GluR puncta, resulting in a severe reduction in the sum intensity of each GluR subunit in *hiw* mutants (Fig. 1e–h and Supplementary Fig. 2a–e). Similar changes in GluRIIA- and GluRIIB-containing receptors were observed in wnd-OE (Fig. 1e–h and Supplementary Fig. 2a–e).

This reduction in GluR levels could result from a regulated change in receptors specifically, or be a consequence of a more general remodeling of the postsynaptic density. We labeled the postsynaptic density with Dlg, a homolog of the postsynaptic scaffold PSD-95, and observed a disorganized structure, including ~60% reduction in intensity in *hiw* and wnd-OE compared to wild type (Fig. 1e, i and Supplementary Fig. 2a). However, GluR puncta number and intensity levels, as well as Dlg organization were restored to wild-type values in *hiw;wnd* double mutants (Fig. 1e–I and Supplementary Fig. 2a–e), consistent with the restoration of mEPSP amplitude in this condition. Thus, activation of Wnd signaling in neurons, either through loss of *hiw* or overexpression of *wnd*, leads to a downregulation of GluR abundance and remodeling of the postsynaptic scaffold in the muscle.

Next, we tested the requirements of *hiw* and *wnd* expression in each synaptic compartment at the NMJ. First, we expressed *hiw* in neurons in a *hiw*-mutant background. As previously reported, this fully rescued the exuberant synaptic growth and reduced presynaptic vesicle intensity levels characteristic of *hiw* mutants[25] (Supplementary Fig. 3c,d). Further, EPSP amplitude, mEPSP amplitude, and postsynaptic GluR levels were all restored to wild-type levels in this condition (Supplementary Fig. 3a,b,e,f). In contrast, muscle expression of *hiw* in *hiw* mutants did not rescue bouton numbers, synaptic vesicle intensity levels, neurotransmission, or glutamate receptor levels (Supplementary Fig. 3a–f). Finally, overexpression of *wnd* in the muscle had no significant impact on synaptic growth, structure, GluR levels, or neurotransmission (Supplementary Fig. 3a–f). Thus, neuronal Wnd signaling transforms the muscle into an altered state of reduced sensitivity to neurotransmitter through a reduction in GluR abundance.

**Acute activation of Wnd signaling reduces GluRs**. Dramatic changes in presynaptic growth and structure are induced when Wnd signaling is chronically activated throughout larval development[10] (Supplementary Fig. 1). This raises a question about whether the postsynaptic remodeling of GluRs and Dlg are a specific response to presynaptic Wnd signaling, or rather are confounded by developmental anomalies of synaptogenesis and maturation occurring in parallel with constitutive Wnd signaling. We therefore sought to activate Wnd signaling at third instar larval stages, a developmental time at which synaptogenesis, growth, and maturation has been largely established[34]. To acutely induce Wnd signaling, we took advantage of the GeneSwitch system, which enables temporal control of Gal4-mediated gene expression through exogenous application of the drug RU486[35]. In particular, we generated larvae with the neuronal GeneSwitch driver *elav-GS* in combination with *UAS-wnd* and fed early third instar larvae RU486 for 24, 48, or 72 h to acutely induce *wnd* expression and assay NMJ structure and function (schematized in Fig. 2a).

Control larvae (*elav-GS>UAS-wnd*) raised in the absence of RU486 showed no significant differences in synaptic growth, function, or GluR levels compared with wild type (Supplementary Table 1), demonstrating no "leakiness" of *wnd* expression in absence of the drug. Exposure of larvae to RU486 for an extended period of 72 h induced the structural hallmarks of Wnd signaling, including exuberant bouton number, increased neuronal surface area, and diminished vGlut intensity (Fig. 2b, c). However, while presynaptic growth and structure remained similar to wild type following 24 and 48 h exposure to RU486, mEPSP and EPSP amplitudes were significantly reduced to levels observed during chronic Wnd signaling (Fig. 2d, e), with a concomitant reduction in GluR levels (Fig. 2f, g). This underscores two important points. First, the postsynaptic response to presynaptic Wnd signaling can be uncoupled from synaptic overgrowth and aberrant structure in the presynaptic compartment. Second, the earliest consequences of presynaptic Wnd signaling include postsynaptic remodeling of GluR fields.

Although overall presynaptic neurotransmitter release is not significantly impacted by presynaptic Wnd signaling, we considered whether this process may mediate an activity-dependent anterograde signal to induce postsynaptic remodeling of GluRs. We expressed tetanus toxin (TNT) in the neurons innervating muscles 6/7 and 4 using the *OK319-Gal4* driver. TNT cleaves the vesicular SNARE protein n-Syb and blocks all evoked neurotransmitter release at the *Drosophila* NMJ[36]. TNT expression in wild type, *hiw*, or wnd-OE led to a complete block of evoked neurotransmission without affecting mEPSP amplitudes, as expected (Fig. 3a, b). Interestingly, the reduced mEPSP amplitudes and GluR levels characteristic of Wnd signaling, as well as the exuberant synaptic growth, persisted in *hiw*+TNT and wnd-OE+TNT (Fig. 3c–f). Thus, presynaptic Wnd signaling converts the postsynaptic muscle into a state of diminished responsivity to neurotransmitter independently of action potential-induced synaptic activity.

**NMJs fail to express PHP following Wnd signaling**. Typically, a reduction in postsynaptic GluR levels activates an adaptive retrograde signaling system in the muscle that induces the expression of PHP[27,37,38]. PHP is defined by increased presynaptic neurotransmitter release (quantal content) following a decrease in postsynaptic receptivity to neurotransmitter (mEPSP amplitude). However, while postsynaptic GluRs and mEPSP amplitudes are reduced following presynaptic Wnd signaling, we observed no change in quantal content, resulting in a seemingly maladaptive reduction in EPSP amplitude (synaptic strength). We therefore considered whether synapses experiencing Wnd signaling are capable of expressing PHP when further challenged by pharmacological or genetic perturbations to the already diminished postsynaptic GluRs.

We first asked whether PHP can be induced acutely and expressed at NMJs experiencing neuronal Wnd signaling. At wild-type NMJs, we confirmed that application of Philanthotoxin-433 (PhTx), an irreversible antagonist of GluRIIA-containing GluRs, results in a ~50% reduction in mEPSP amplitude and a robust increase in quantal content that maintains baseline EPSP amplitude (Fig. 4a–d). Application of PhTx to *hiw*-mutant and wnd-OE synapses further reduced mEPSP amplitudes below the already reduced level (Fig. 4a, b). However, EPSP amplitudes were also further reduced following PhTx application (Fig. 4a, c), and no increase in presynaptic release (quantal content) was observed (Fig. 4a, d). Importantly, PHP also fails to be expressed at NMJs in which Wnd signaling is acutely induced for 48 h (Supplementary Fig. 4), indicating that the inhibition of PHP signaling occurs in parallel to the postsynaptic response to presynaptic Wnd signaling. In contrast, PHP was robustly expressed in *hiw;wnd* double mutants following PhTx application (Fig. 4a–d). Together, this

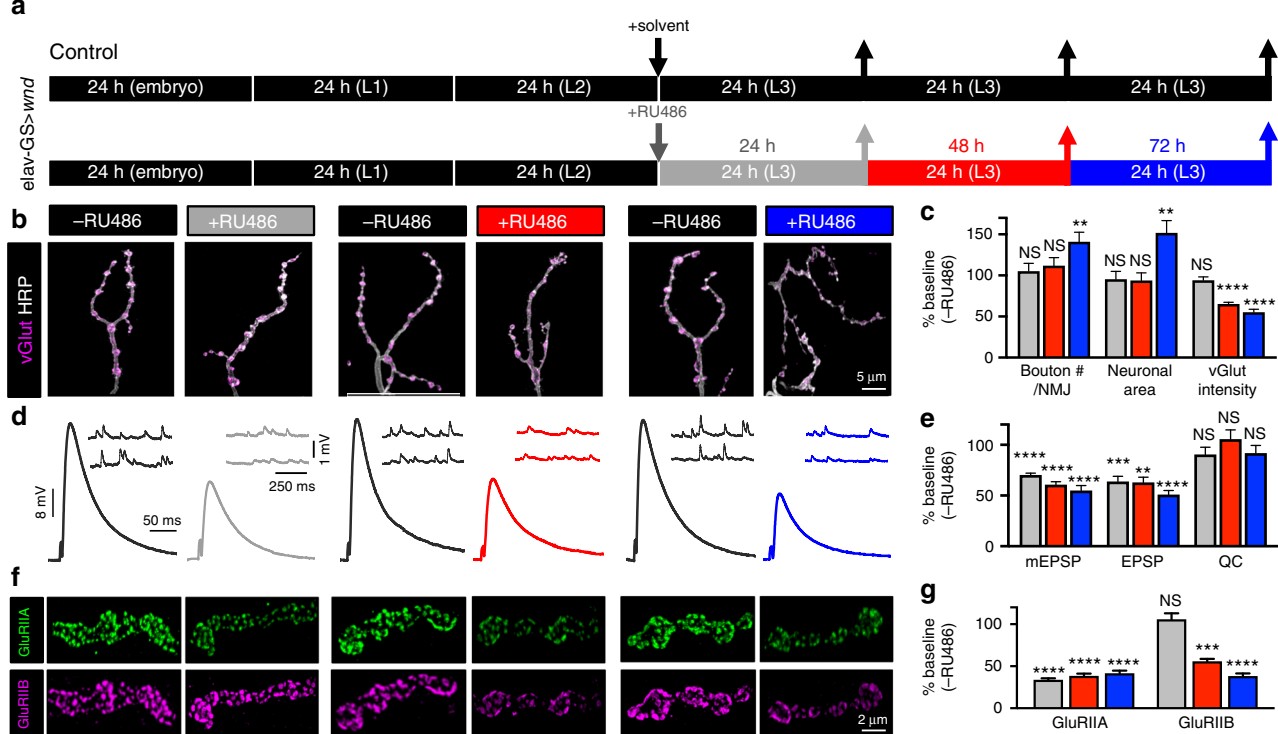

**Fig. 2** Acute activation of neuronal *wnd* signaling rapidly induces GluR remodeling without impacting presynaptic growth or structure. **a** Schematic illustrating the temporal administration of RU486 to acutely induce neuronal *wnd* expression in late stages of larval development. Control genotypes (*UAS-wnd*/+; *elav-GS*/+) were reared in the same conditions but fed saline without RU486, while experimental animals where fed RU486 at the indicated time points. Staining and electrophysiological experiments were performed after 24 h (gray), 48 h (red), and 72 h (blue) following exposure to RU486. **b** Representative images of NMJs immunostained with antibodies that recognize the synaptic vesicle marker vGlut (vesicular glutamate transporter; magenta) and the neuronal membrane marker HRP (white). **c** Quantification of total bouton number per NMJ, neuronal membrane surface area, and vGlut intensity levels normalized to control (larvae at the same time points raised on fly food without RU486). **d** Representative mEPSP and EPSP traces from NMJs in the indicated genotypes and conditions. **e** Acute activation of neuronal *wnd* signaling induces reductions in mEPSP and EPSP amplitudes, but no change in quantal content. **f** Representative images NMJs in the indicated conditions immunostained with antibodies against two postsynaptic glutamate receptor subunits (GluRIIA and GluRIIB). **g** Quantification of total fluorescence intensity of each GluRIIA or GluRIIB puncta normalized to baseline (same time point without RU486) reveals acute activation of neuronal *wnd* signaling rapidly remodels postsynaptic GluR levels. Error bars indicate ± SEM. One-way analysis of variance (ANOVA) test was performed, followed by a Tukey's multiple-comparison test. *$p$ 0.05; **$p$ 0.01; ***$p$ 0.001; ****$p = 0.0001$, NS not significant, $p > 0.05$

demonstrates that the postsynaptic induction and/or presynaptic expression mechanisms necessary for acute PHP signaling is disrupted at synapses experiencing neuronal Wnd signaling.

Wnd is chronically activated in *hiw* mutants and wnd-OE, while PHP is not induced following acute block of GluRs. Differences have been demonstrated between acute and chronic forms of PHP induction[39], so it is possible that if GluRs were disrupted over longer timescales, PHP signaling may be induced and expressed. To assess this possibility, we utilized a genetic mutation in the postsynaptic GluR subunit *GluRIIA* (*GluRIIA*). In this mutant, mEPSP amplitudes are reduced throughout development but normal EPSP amplitudes are preserved due to a chronic, homeostatic increase in quantal content[38,40] (Fig. 5a–d). Similar to PhTx application, however, *hiw;GluRIIA* mutants also exhibited further reduced mEPSP amplitudes but failed to express a compensatory increase in presynaptic release compared to *hiw* mutants alone, resulting in reduced EPSP amplitude (Fig. 5a–d). Thus, activation of Wnd signaling in neurons prevents the induction and/or expression of PHP over both acute and chronic timescales, even following an additional reduction in postsynaptic GluR activity.

**Retrograde homeostatic signaling is silenced in *hiw* mutants.** The failure of *hiw*-mutant synapses to express PHP could be due

to an inability to increase presynaptic neurotransmitter release in response to retrograde signaling from the postsynaptic cell. Alternatively, retrograde signaling from the postsynaptic cell may fail to be communicated in *hiw*-mutant synapses despite diminished GluR function. In this scenario, PHP expression would be blocked because of a failure to receive a retrograde signal from the postsynaptic cell, rather than an inability to homeostatically modulate presynaptic efficacy. Although little is known about the postsynaptic signaling system necessary to drive the induction of PHP, overexpression of the translational regulator Target of Rapamycin (Tor-OE) in muscle triggers PHP signaling, even in the absence of any perturbation to GluRs[39,41,42]. Tor-OE was interpreted to artificially drive a change in protein synthesis necessary to activate retrograde PHP signaling in the absence of the signaling that normally initiates this process following GluR perturbation. We next tested whether *hiw* mutants are competent to express PHP when emission of the retrograde signal is forced from the muscle by Tor-OE.

Postsynaptic overexpression of Tor (Tor-OE) at wild-type NMJs led to the expected increase in EPSP amplitude and quantal content, with no change in mEPSP amplitude (Fig. 6a–e). Further, PhTx application to Tor-OE diminished mEPSP amplitudes as expected, but no change in quantal content was observed because PHP was already fully activated by Tor-OE[39] (Fig. 6a–e). Surprisingly, Tor-OE in *hiw* mutants (*hiw*+Tor-OE) resulted in

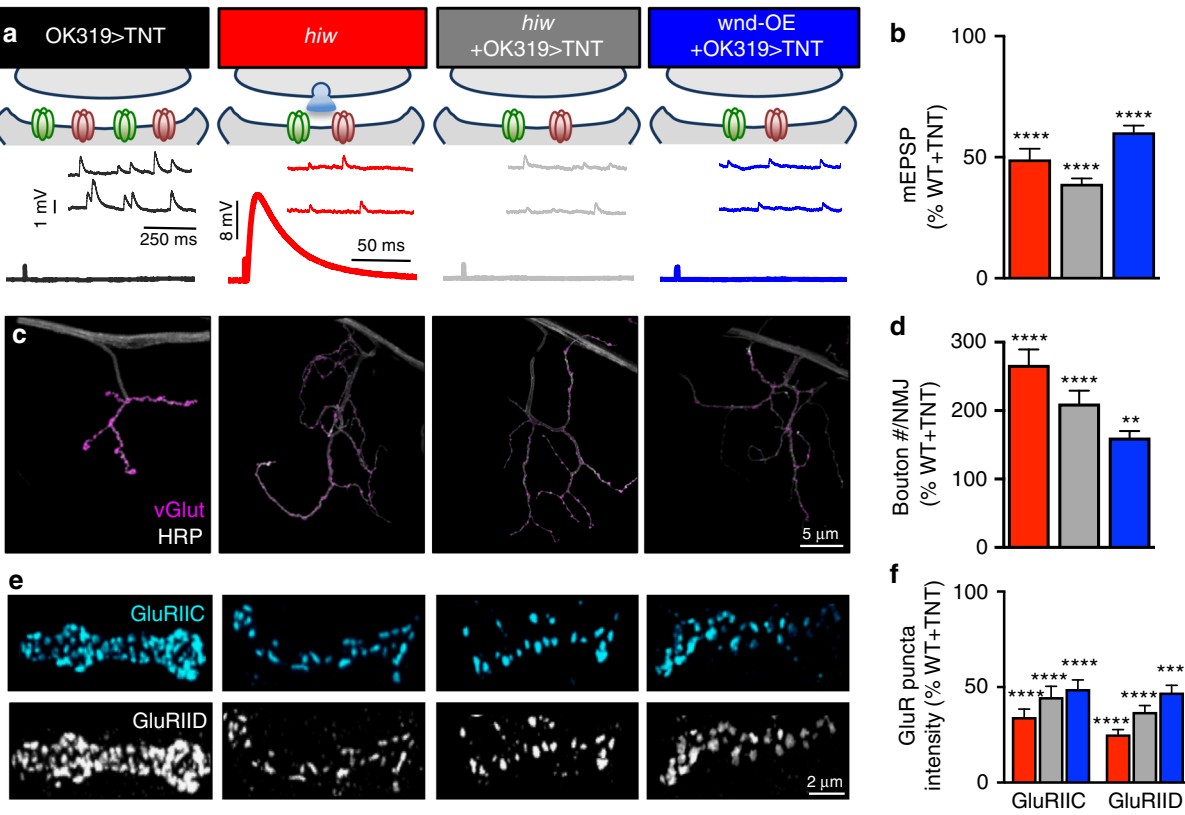

**Fig. 3** Presynaptic Wnd signaling induces GluR remodeling independently of evoked neurotransmitter release. **a** Schematic and representative traces of NMJs expressing tetanus toxin in otherwise wild type backgrounds (OK319>TNT: *w;OK319-Gal4/+;UAS-TNT/+*), in *hiw* mutants (*hiw*+OK319>TNT: *hiw^ΔN;OK319-Gal4/+;UAS-TNT/+*), and in wnd-OE (wnd-OE+OK319>TNT: *w;OK319-Gal4/UAS-wnd;UAS-TNT/+*). **b** Quantification of mEPSP amplitude in the indicated genotypes normalized to OK319>TNT values. Note that reduced mEPSP values persist in *hiw* and wnd-OE despite the absence of evoked activity. Representative NMJs of the indicated genotypes immunostained (**c**) and quantified (**d**) with antibodies against the synaptic vesicle marker vGlut (magenta) and neuronal membrane marker HRP (white). The exuberant synaptic overgrowth characteristic of *hiw* and wnd-OE also persists in the absence of evoked activity. **e** Representative images of boutons immunostained with antibodies against the common postsynaptic glutamate receptor subunits GluRIIC and GluRIID. **f** Quantification of individual GluR puncta intensity normalized to OK319>TNT. Receptor levels in *hiw* and wnd-OE are reduced despite the absence of evoked activity. Error bars indicate ± SEM. One-way analysis of variance (ANOVA) test was performed, followed by a Tukey's multiple-comparison test. **p 0.001; ****p 0.0001

a similar increase in presynaptic neurotransmitter release and EPSP amplitude (Fig. 6a, c), demonstrating that *hiw* synapses can still express PHP when retrograde homeostatic signaling is transduced in the muscle (Fig. 6d, e). Further, PhTx application to *hiw*+Tor-OE also did not increase quantal content (Fig. 6a–e), consistent with conventional retrograde PHP signaling being activated in this condition. Interestingly, we observed increased mEPSP amplitudes in *hiw*+Tor-OE compared to *hiw* mutants alone (Fig. 6a, b), which was reflected by an increase in both GluRIIA- and GluRIIB-containing receptors (Fig. 6f, g). This result was unexpected, as Tor-OE in wild-type synapses has no effect on mEPSP amplitude or GluR levels[42] (Fig. 6b, f, g). Tor-OE drives a global increase in muscle protein synthesis[41], which may contribute to the enhanced GluR levels in *hiw* and enhanced Dlg levels in both Tor-OE and *hiw*+Tor-OE (Fig. 6f, g). Importantly, the enhanced quantal content observed in *hiw*+Tor-OE occurred in addition to and above the increase in mEPSP amplitude (Fig. 6e), demonstrating that PHP was indeed induced and expressed in *hiw*+Tor-OE.

It is possible that overexpression of Tor in the muscle somehow suppressed neuronal Wnd signaling, and that this suppression, rather than PHP signaling, increased presynaptic neurotransmitter release. To assess this possibility, we examined synaptic growth, structure, and protein levels in *hiw*+Tor-OE compared with controls. First, we confirmed that Tor-OE does not change synapse morphology, growth, or levels of synaptic proteins compared with wild type (Supplementary Fig. 5a–c). Further, *hiw*-mutant NMJs exhibited the characteristic expansion of presynaptic terminals and reduction in presynaptic protein levels, and this was not altered by Tor-OE (Supplementary Fig. 5a–c). Finally, the puc-lacZ reporter, which sensitively tracks transcriptional responses to Wnd/DLK signaling in the nucleus[9], was enhanced in *hiw* neurons, as expected, and was similarly elevated in *hiw*+Tor-OE (Supplementary Fig. 5a,d). Thus, we find no evidence that postsynaptic Tor-OE alters presynaptic Wnd signaling, consistent with Tor-OE specifically driving PHP-related trans-synaptic communication.

Together, these results indicate that Tor-OE exerts two distinct impacts on the muscle in *hiw* mutants: One is to trigger retrograde PHP signaling as it does in wild type. The second is to increase GluR levels, an effect that is not observed in wild type. The restoration of PHP expression by Tor-OE demonstrates that neurons with active Wnd signaling retain the capacity to homeostatically modulate presynaptic release in response to retrograde communication from the postsynaptic compartment, and suggests that the muscle in *hiw* mutants does not emit this signal despite reduced GluR abundance, even after additional inhibitions to GluR function.

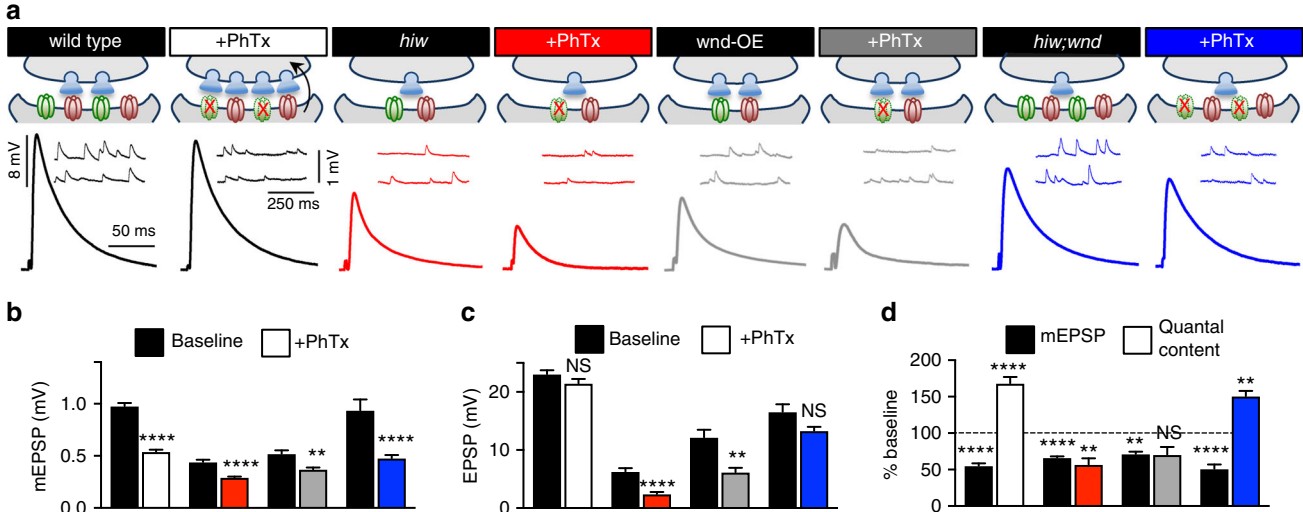

**Fig. 4** PHP is not expressed in *hiw* mutants or wnd-OE following acute postsynaptic GluR perturbation. **a** Schematic and representative EPSP and mEPSP traces of indicated genotypes before (baseline) and following PhTx application. Quantification of mEPSP (**b**) and EPSP (**c**) amplitude in the indicated genotypes and conditions. Note that while PhTx application causes a reduction in mEPSP amplitudes in all genotypes, only *hiw* mutants and wnd-OE fail to maintain baseline EPSP amplitudes. **d** Quantification of mEPSP and quantal content values normalized to the same genotype in the absence of PhTx (baseline). No increase in presynaptic neurotransmitter release (quantal content) is observed in *hiw*+PhTx or wnd-OE+PhTx, demonstrating that PHP fails to be acutely expressed in *hiw* mutants, but is restored in *hiw;wnd* double mutants. Error bars indicate ± SEM. For statistical significance, an unpaired *t*-test was performed between respective genotypes with and without PhTx. **p 0.01; ****p 0.0001; NS not significant, p > 0.05

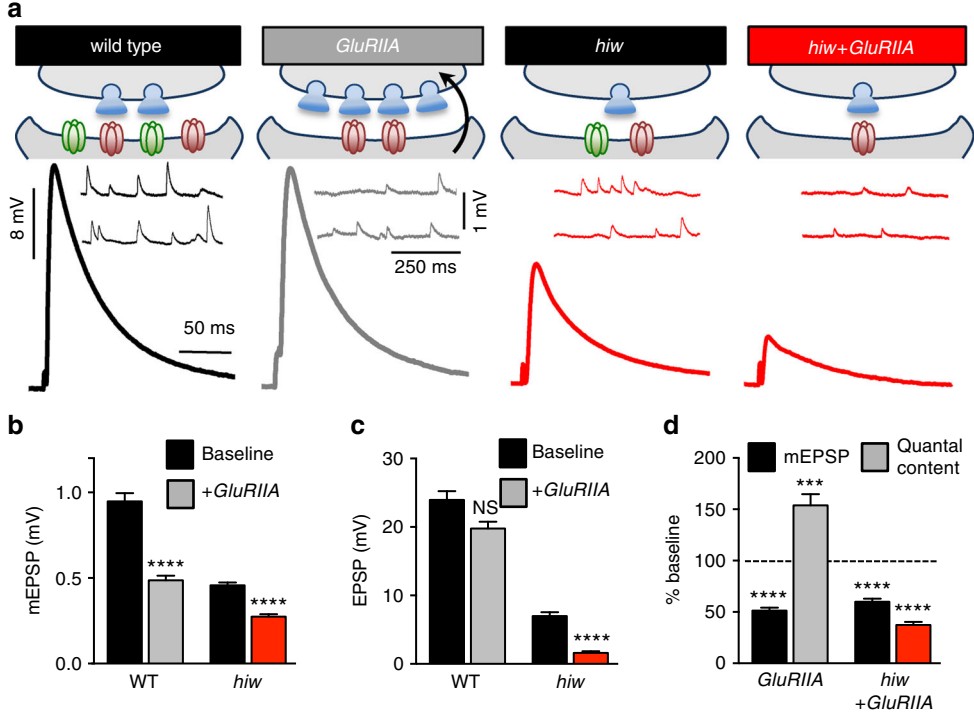

**Fig. 5** *hiw* mutants fail to express PHP following chronic loss-of-*GluRIIA* receptors. **a** Schematic and representative EPSP and mEPSP traces of recordings in wild type, *GluRIIA* mutants (*GluRIIA*: w;GluRIIA$^{SP16}$), *hiw* mutants, and *hiw;GluRIIA* double mutants (hiw$^{ΔN}$;GluRIIA$^{SP16}$). Quantification of mEPSP (**b**) and EPSP (**c**) amplitude in the indicated genotypes shows a reduction in mEPSP amplitude but failure to maintain baseline EPSP levels in *hiw;GluRIIA*. **d** Quantification of mEPSP and quantal content values normalized to baseline (without loss-of-*GluRIIA*) of the same genotype. No homeostatic increase in quantal content is observed in *hiw;GluRIIA* mutants. Error bars indicate ± SEM. For statistical significance, an unpaired *t*-test was performed. ***p 0.001; ****p 0.0001; NS not significant, p > 0.05

**Increased GluR expression fails to restore PHP in *hiw* mutants.** Tor-OE, in addition to triggering PHP signaling and expression in *hiw* mutants, also partially restored GluR expression. We considered the possibility that perhaps restoration of GluR levels alone is sufficient to confer competence to transduce postsynaptic PHP signaling in *hiw* mutants. To address this, we overexpressed the *GluRIIA* subunit in the muscle of *hiw* mutants (hiw+GluR-IIA-OE). GluRIIA-OE in wild type increases the synaptic

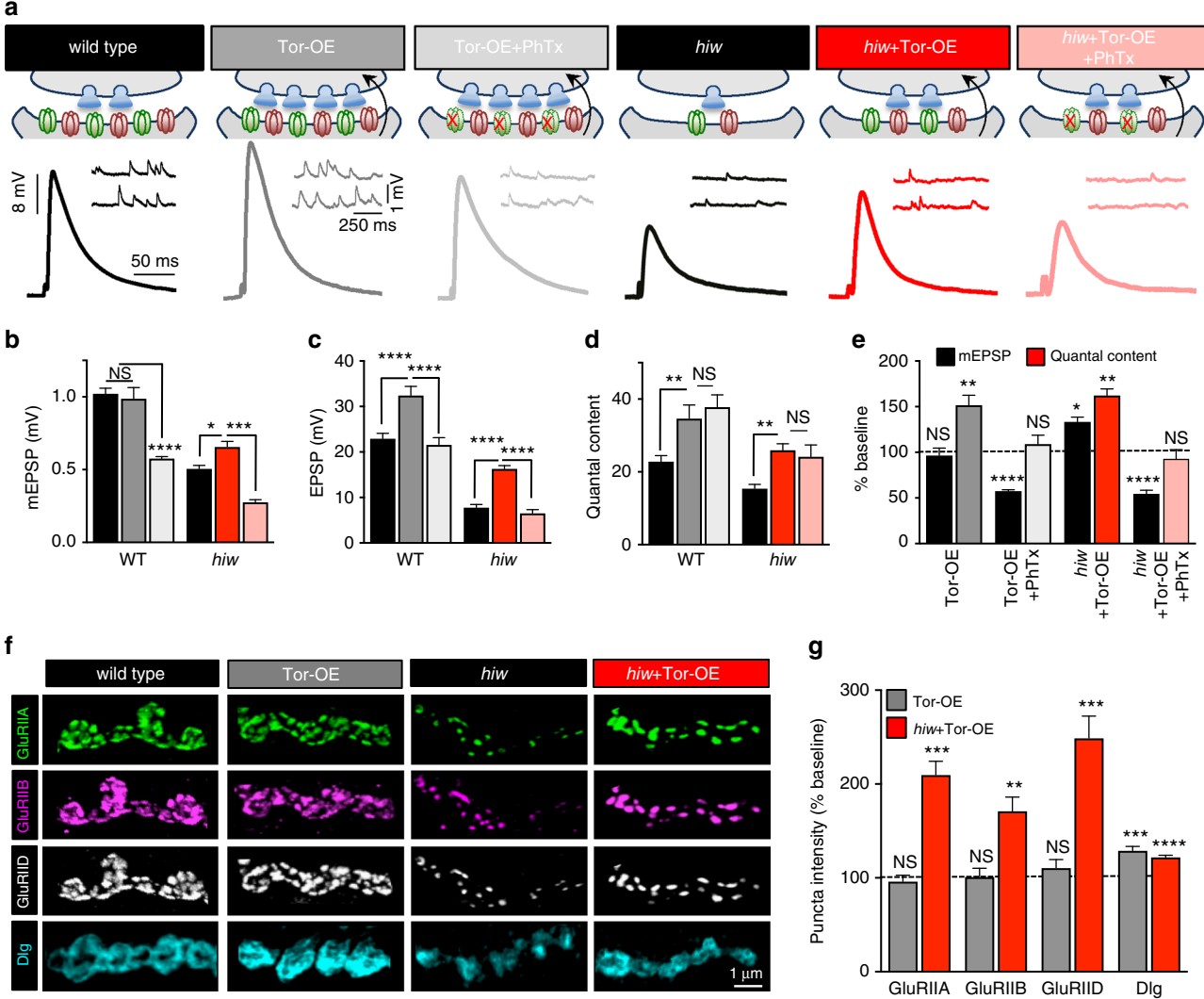

**Fig. 6** *hiw*-mutant NMJs can express PHP when retrograde signaling is induced through postsynaptic Tor overexpression. **a** Schematic illustrating postsynaptic overexpression of Tor at wild type (Tor-OE; *w;MHC-Gal4/UAS-Tor*) and in *hiw* mutants (*hiw*+Tor-OE: *hiw^ΔN;MHC-Gal4/UAS-Tor*), which leads to an increase in presynaptic neurotransmitter release in each condition. Application of PhTx to *hiw*+Tor-OE synapses reduces mEPSP amplitude but presynaptic release is unchanged compared to the *hiw*+Tor-OE condition. Representative mEPSP and EPSP traces of the indicated genotypes are shown below. Quantification of mEPSP amplitude (**b**), EPSP amplitude (**c**), and quantal content (**d**) in the indicated genotypes. An increase in mEPSP amplitude is observed in *hiw*+Tor-OE, in contrast to Tor-OE at wild-type NMJs. However, EPSP amplitude and quantal content are still increased in both conditions. **e** Quantification of mEPSP and quantal content values normalized to the same genotype in the absence of Tor-OE. PhTx application to Tor-OE and *hiw* +Tor-OE reduces mEPSP and EPSP amplitude without changing quantal content compared to their baseline (no PhTx application) values. Representative images (**f**) and quantification (**g**) of postsynaptic GluR levels at NMJs of the indicated genotypes immunostained with antibodies that recognize GluRIIA, GluRIIB, and GluRIID subunits, and Dlg. Note that Tor-OE in *hiw* mutants results in increased expression of all postsynaptic receptor subunits, whereas receptors are unchanged when Tor is overexpressed in wild-type muscle. An increase in Dlg intensity is observed when Tor is overexpressed at wild-type and *hiw* synapses. Error bars indicate ± SEM. For statistical significance, an unpaired *t*-test was performed between respective genotypes with and without Tor-OE. *$p \leq 0.05$; **$p \ 0.01$; ***$p \ 0.001$; ****$p \ 0.0001$; NS not significant, $p > 0.05$

expression of GluRIIA-containing receptors and Dlg, while decreasing that of GluRIIB-containing receptors[40], suggesting competition between A- and B-type receptors at postsynaptic densities (Fig. 7a, b and Supplementary Fig. 6a–g). However, GluRIIA-OE in *hiw* mutants (*hiw*+GluRIIA-OE) actually increased levels of both GluRIIA- and GluRIIB-containing receptors at synapses, as well as Dlg (Fig. 7a, b and Supplementary Fig. 6a–g). This indicates that *GluRIIA* expression may be limiting in *hiw*-mutant muscle.

We next performed an electrophysiological analysis of GluRIIA-OE and *hiw*+GluRIIA-OE. In control NMJs, GluRIIA-OE resulted in an increase in mEPSP amplitude and a corresponding increase in EPSP amplitude, but no change in

quantal content, consistent with previous studies[38,40] (Fig. 7c–g). We also observed a similar increase in both mEPSP and EPSP amplitude in *hiw*+GluRIIA-OE without a change in quantal content (Fig. 7c–g), consistent with the results of GluR staining in this condition. Finally, we applied PhTx to GluRIIA-OE and *hiw* +GluRIIA-OE NMJs, which resulted in a large reduction in mEPSP amplitude in both genotypes (Fig. 7c, d). However, while GluRIIA-OE+PhTx displayed a robust homeostatic increase in quantal content, no increase was observed in *hiw*+GluRIIA-OE +PhTx (Fig. 7c–g). Thus, *hiw* mutants are incapable of initiating homeostatic retrograde signaling following perturbations to GluR functionality, even when the expression of GluRs is restored to near wild-type levels. This suggests that *hiw*-mutant synapses

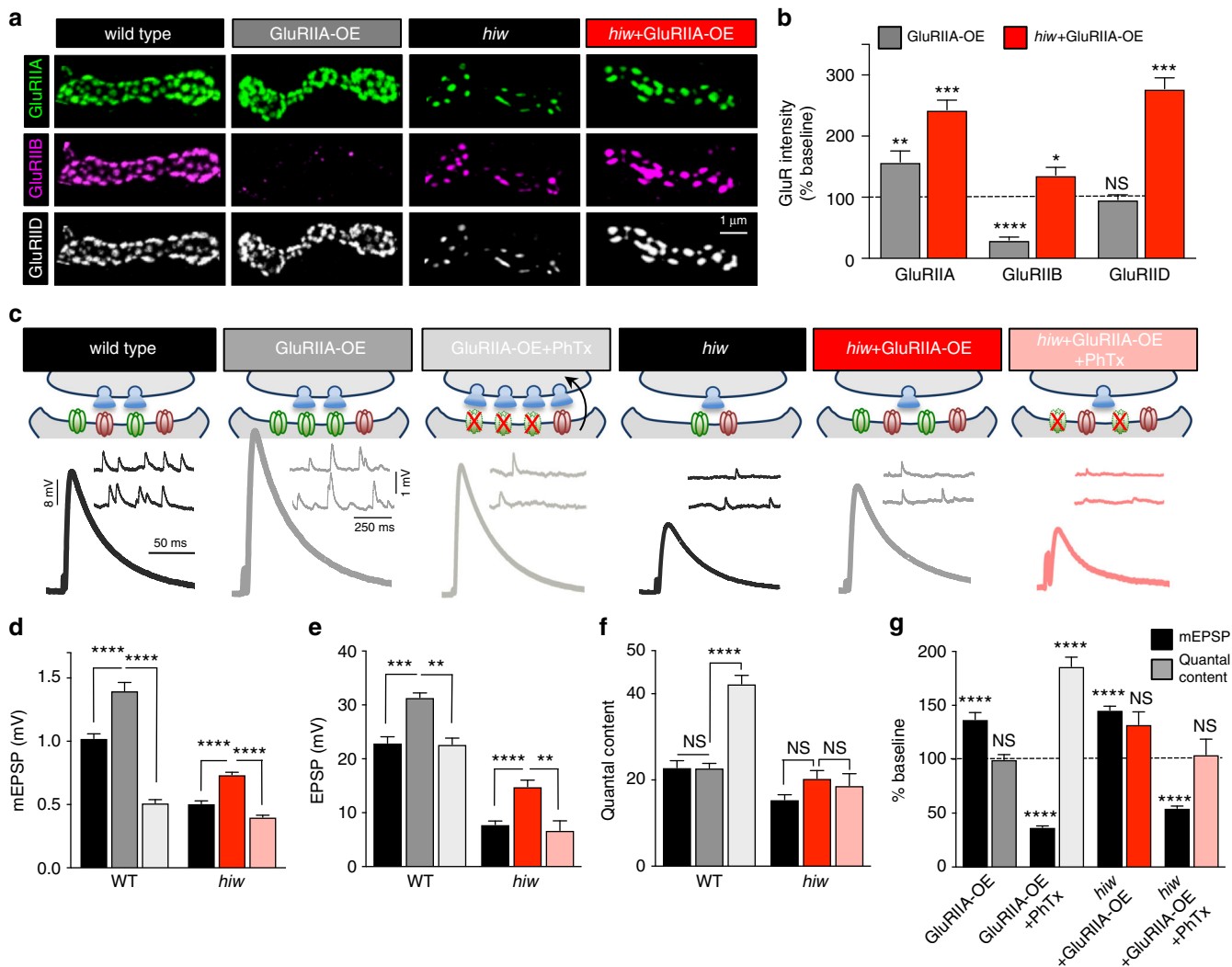

**Fig. 7** Increased GluR expression does not restore PHP signaling in *hiw*-mutant muscle. Representative images (**a**) and quantification (**b**) of postsynaptic GluR levels at wild-type, postsynaptic overexpression of *GluRIIA* (GluRIIA-OE; *w;MHC-Gal4/UAS-dGluRIIA*), *hiw*-mutant, and *hiw*+GluRIIA-OE (*hiw*+GluRIIA-OE: *hiw^{ΔN};MHC-Gal4/UAS-dGluRIIA*) NMJs. GluRIIA-OE in wild-type NMJs results in a reduction in GluRIIB subunits, leaving GluRIID levels unchanged, while all glutamate receptor subunits are increased in *hiw*+GluRIIA-OE compared to *hiw* mutants alone. **c** Schematic illustrating postsynaptic overexpression of *GluRIIA* in wild type and *hiw* mutants at baseline and following PhTx application. Application of PhTx to GluRIIA-OE NMJs leads to a homeostatic increase in presynaptic release, while this increase is not observed in *hiw*+GluRIIA-OE NMJs. Representative mEPSP and EPSP traces of the indicated genotypes are shown below. Quantification of mEPSP amplitude (**d**), EPSP amplitude (**e**), and quantal content (**f**) in the indicated genotypes. **g** Quantification of mEPSP and quantal content values normalized to baseline (without GluRIIA-OE) in the indicated genotypes. For GluRIIA-OE+PhTx and *hiw*+GluRIIA-OE+PhTx, values are normalized to GluRIIA-OE and *hiw*+GluRIIA-OE, respectively. Error bars indicate ± SEM. For statistical significance, an unpaired *t*-test was performed between respective genotypes with and without GluRIIA-OE. *$p \leq 0.05$; **$p$ 0.01; ***$p$ 0.001; ****$p$ 0.0001; NS not significant, $p > 0.05$

disrupt PHP signaling downstream of GluR perturbation but upstream of Tor-dependent mechanisms.

**Lowered synaptic strength is homeostatically stabilized in *hiw*.**
Thus far, our data demonstrate that neurons experiencing Wnd signaling still have the capacity to express PHP in response to retrograde communication from the muscle. However, the post-synaptic muscle in this condition is transformed into a state that disrupts the signal transduction system that normally operates to transduce retrograde PHP communication when GluRs are perturbed. One possibility is that all forms of homeostatic signaling are disrupted, and the observed reduction in synaptic strength is simply a result of reduced GluR levels. Alternatively, synaptic strength may still be under homeostatic control, but recalibrated

at a reduced level, where the loss-of-PHP signaling may be necessary to prevent synaptic strength from adjusting to wild-type levels following reduced GluR expression. In this model, the specific disruption of PHP signaling would be a necessary adaptation to both prevent retrograde communication and maintain synaptic strength at diminished levels, while distinct homeostatic mechanisms may still operate to stabilize this reduced synaptic state.

We sought to distinguish between these possibilities and examine whether other homeostatic mechanisms are operational in muscle responding to neuronal injury-related signaling. We turned our attention to a distinct homeostatic mechanism that has been observed at the *Drosophila* NMJ, where hypo-innervation decreases presynaptic neurotransmitter release in proportion to the reduction in synapse number, but synaptic

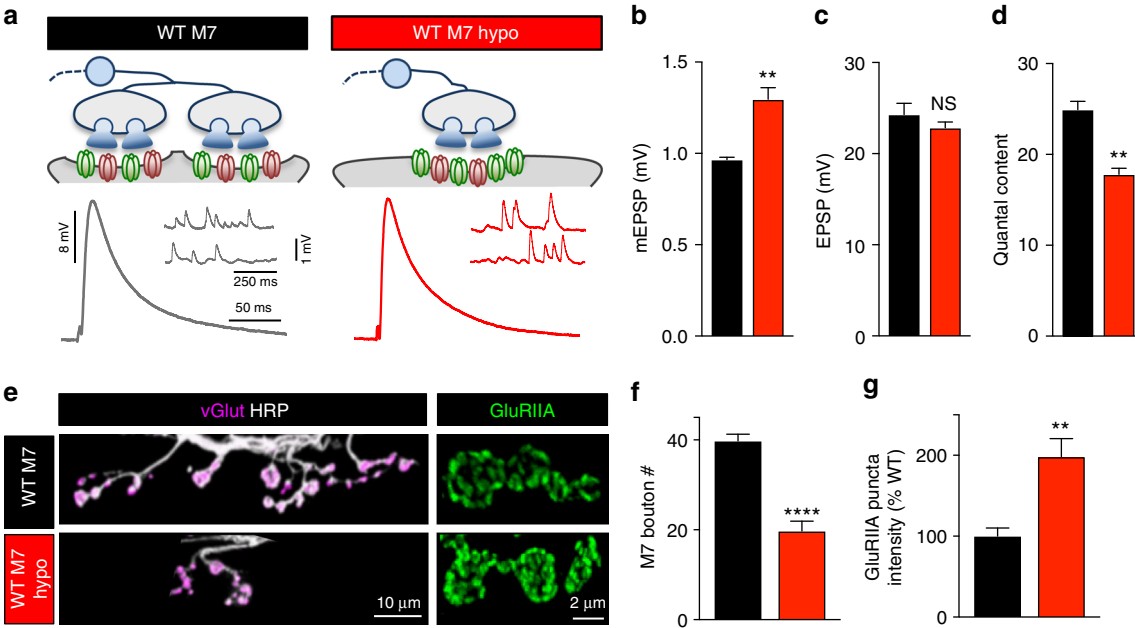

**Fig. 8** Hypo-innervation induces a homeostatic increase in postsynaptic GluR abundance. **a** Schematic illustrating a wild type and hypo-innervated muscle 7 synapse at the *Drosophila* NMJ achieved by overexpression of *FasII* selectively on the adjacent muscle 6 (WT M7 hypo: *w;tubP>stop>Gal4,UAS-FLP,UAS-CD8-GFP/UAS-FasII-PEST+;H94-Gal4,nSyb-Gal80*). A compensatory increase in mEPSP amplitude is observed that maintains baseline levels of synaptic strength (EPSP amplitude). Representative mEPSP and EPSP traces of the indicated genotypes recorded are shown below. Quantification of mEPSP amplitude (**b**), EPSP amplitude (**c**), and quantal content (**d**) in the indicated genotypes. **e** Representative images of muscle 7 larval NMJs in wild type (WT) and WT M7 hypo immunostained with antibodies that recognize vGlut and the neuronal membrane (HRP). Note the reduced innervation of boutons onto muscle 7 in the WT M7 hypo condition. Right: Representative GluRIIA staining of the corresponding genotypes. An upregulation in GluRIIA levels is observed on the hypo-innervated muscle 7. Quantification of bouton numbers (**f**) and GluRIIA puncta intensity (**g**) of the indicated genotypes. Error bars indicate ± SEM. For statistical significance, an unpaired *t*-test was performed. **$p$ 0.01; ****$p$ 0.0001; NS not significant, $p > 0.05$

strength is maintained at baseline levels due to a homeostatic increase in the postsynaptic responsiveness to neurotransmitter[30]. Specifically, hypo-innervation of the muscle 7 NMJ is achieved by overexpression of the cell adhesion factor *Fascilin II* (*FasII*) selectively on the adjacent muscle 6[30] (Fig. 8a and Methods). This biased expression of *FasII* recruits additional innervation on muscle 6 at the expense of muscle 7, resulting in reduced presynaptic neurotransmitter release at muscle 7 (Fig. 8a–d). However, synaptic strength is maintained at baseline values due to a compensatory increase in mEPSP amplitude[30], which we confirmed (Fig. 8a–c). The expression mechanism for the change in quantal size has not been defined, so we sought to characterize this form of homeostatic plasticity in more detail.

We first repeated this manipulation, improving the expression of *FasII* on muscle 6 (WT M7 hypo; Methods section). We confirmed a ~50% reduction in bouton number on muscle 7 in M7 hypo compared to wild type (Fig. 8e, f). The enhanced mEPSP amplitude could be due to a change in the amount of glutamate released in each synaptic vesicle (a presynaptic expression mechanism), or alternatively, an increase in post-synaptic GluR levels. We immunostained GluRs and found a ~200% increase in the sum intensity of GluRIIA receptor puncta on M7 hypo compared to wild type (Fig. 8e, g). Therefore, hypo-innervation induces a homeostatic increase in postsynaptic GluRIIA-containing receptor levels that compensates for reduced neurotransmitter release to maintain stable synaptic strength.

Overexpression of *FasII* on muscle 6 of *hiw* mutants (*hiw* M7 hypo) resulted in a similar reduction in bouton number compared to WT M7 hypo (Fig. 9e, f). Interestingly, mEPSP amplitudes on muscle 7 were also significantly increased in *hiw* M7 hypo compared to *hiw* mutants alone, with an increase in GluRIIA-containing receptor levels and no difference in EPSP

amplitude compared with baseline levels (Fig. 9a–g). Together, this data demonstrates two important points (schematized in Fig. 9h). First, at *hiw*-mutant NMJs, the muscle is capable of responding to hypo-innervation by transducing adaptive GluR modulations despite diminished baseline GluR abundance to stabilize synaptic strength. Second, a homeostat, independent of the one controlling PHP, stabilizes the reduced state of synaptic strength induced by neuronal Wnd signaling.

## Discussion

We have characterized the responses of the postsynaptic cell to presynaptic activation of Wnd signaling at the *Drosophila* NMJ. We find that this signaling transforms the postsynaptic target into a state of reduced responsiveness to neurotransmitter release which is stabilized through distinct modulations to two homeostatic plasticity mechanisms (Fig. 9h). These findings define the series of acclimations that occur at synapses following injury-related Wnd signaling, where synaptic strength is recalibrated to a reduced set point by shunting retrograde communication and maintained through a homeostatic GluR scaling mechanism.

The conserved *hiw*/PHR gene family regulates Wnd/DLK signaling and has diverse roles in synaptic development, degeneration and regeneration, and neurotransmission[1,6]. Although there are clearly myriad changes induced in neurons following Wnd activity, at the *Drosophila* NMJ, we have found no evidence for *hiw* or *wnd* having any functions in the postsynaptic muscle. Although Wnd has no apparent function in regulating presynaptic neuro-transmitter release, its negative regulator, Hiw, does have an independent function in promoting neurotransmitter release[10]. The putative substrate that mediates this role for Hiw is unknown, but must be separate from Wnd and the downstream pathways

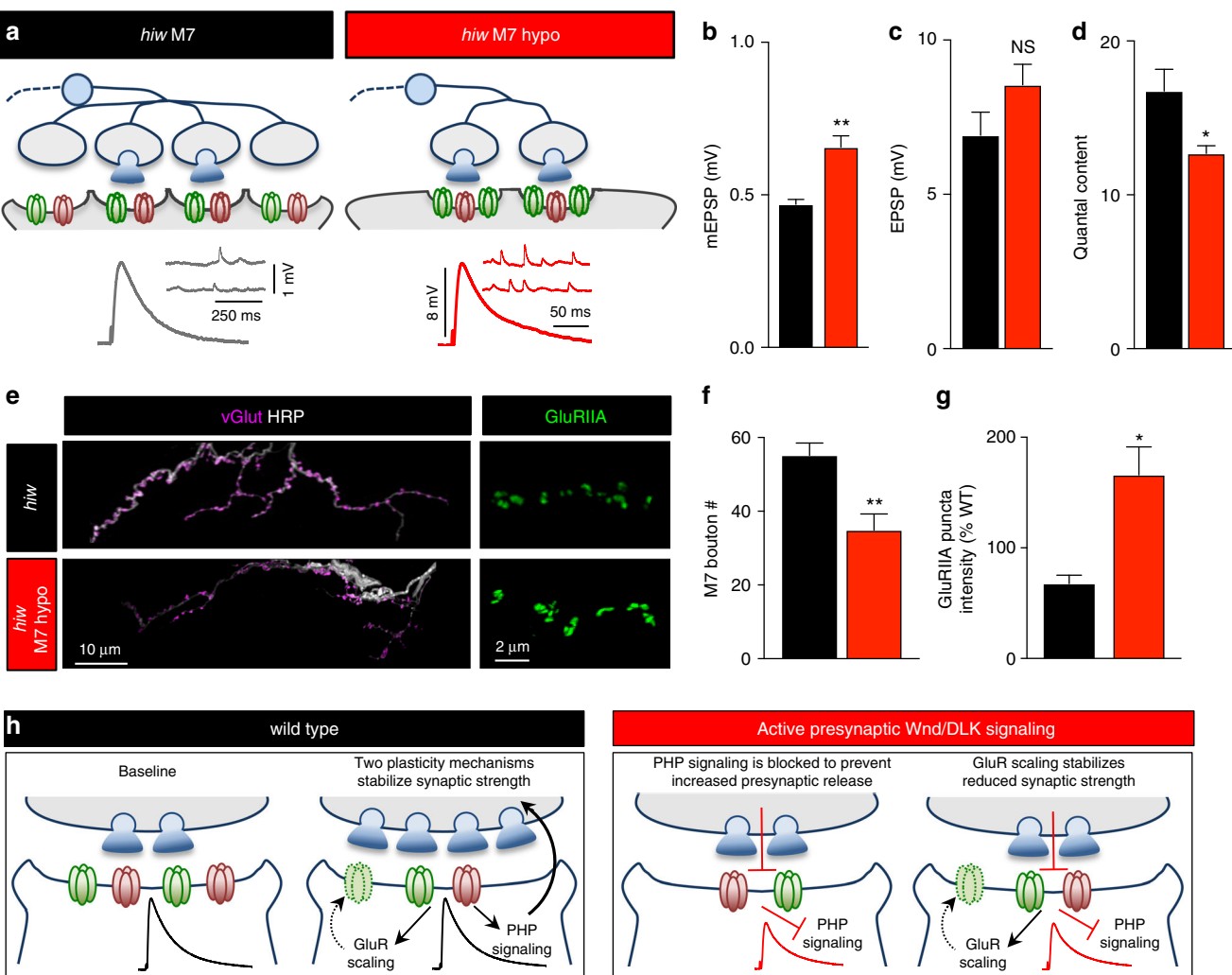

**Fig. 9** Homeostatic control of postsynaptic GluR abundance is induced and expressed following hypo-innervation in *hiw* mutants. **a** Schematic illustrating a *hiw*-mutant NMJ and a hypo-innervated *hiw*-mutant muscle 7 NMJ (*hiw* M7 hypo: *hiw^ΔN*; *tubP>stop>Gal4, UAS-FLP, UAS-CD8-GFP/UAS-FasII-PEST+; H94-Gal4, nSyb-Gal80*). Representative mEPSP and EPSP traces of the indicated genotypes are shown below. Quantification of mEPSP amplitude (**b**), EPSP amplitude (**c**), and quantal content (**d**) in the indicated genotypes demonstrates a homeostatic increase in mEPSP amplitude maintains baseline synaptic strength despite a reduction in quantal content in *hiw* M7 hypo. **e** Representative images of muscle 7 larval NMJs in *hiw* and *hiw* M7 hypo immunostained with antibodies that recognize vGlut and the neuronal membrane (HRP). Note the reduced innervation of boutons onto muscle 7 in the *hiw* M7 hypo condition. Right: Representative GluRIIA staining of the indicated genotypes. An upregulation in GluRIIA levels is observed specifically on *hiw* M7 hypo NMJs. Quantification of bouton numbers (**f**) and GluRIIA puncta intensity (**g**) in the indicated genotypes. **h** Schematic illustrating injury-related adaptations to synaptic function and homeostatic plasticity. Activation of presynaptic DLK signaling induces a reduction in GluR levels. This diminished state of synaptic strength is achieved through a silencing of PHP signaling upstream of Tor activity, and stabilized by GluR scaling. Error bars indicate ± SEM followed by an unpaired *t*-test. *$p \leq 0.05$; **$p\ 0.01$; NS not significant, $p > 0.05$

orchestrated through Wnd signaling. Therefore, following injury, inhibition of Hiw exerts two separate influences that mutes functionality at both synaptic compartments: First, Wnd is activated to provoke a diminished state of responsiveness in the muscle, and second, presynaptic neurotransmitter release is reduced. The substrate that mediates the Wnd-independent role of Hiw to reduce presynaptic efficacy is unknown, but will be an interesting target to define in future studies.

It is remarkable that although Wnd signaling is driven solely in the presynaptic compartment at the NMJ, this process is sensed and transforms the postsynaptic muscle into a novel state characterized by subdued responsiveness to presynaptic activity. Neuronal Wnd signaling must therefore impart anterograde information to the muscle. Importantly, this information is communicated independently of altered synaptic development and structure, as well as evoked neurotransmission. An attractive

possibility is that trans-synaptic adhesion proteins may mediate the activity-independent anterograde signaling following presynaptic Wnd activation. Indeed, multiple extracellular cues and trans-synaptic signals interpose a rich dialog between presynaptic terminals, postsynaptic compartments, and the extracellular matrix at synapses[43–45]. Trans-synaptic "nanocolumns" have recently emerged as inter-cellular signaling complexes that orchestrate synaptic remodeling and plasticity in addition to development, maturation, and structural alignment at synapses[46]. It is tempting to speculate that direct interactions through these nanocolumns may communicate injury-related signaling from presynaptic terminals to postsynaptic partners to remodel receptor fields and plasticity pathways. Intriguingly, there is evidence that many new proteins are expressed in neurons following activation of Wnd/DLK signaling[47], providing possible candidates to test for roles in this process.

One major outcome of the response in muscle to presynaptic Wnd signaling is a diminution of postsynaptic neurotransmitter receptors. This parallels observations at the mouse NMJ following denervation or synapse elimination, in which a reduction in neurotransmitter receptor levels and protein synthesis in muscle has been demonstrated[48–50]. This selective loss-of-acetylcholine receptors at synaptic sites is a result of removal of receptors from these areas coupled with a lack of insertion of new receptors[17–19]. It is likely that similar mechanisms work to reduce GluR levels at the *Drosophila* NMJ following neuronal Wnd signaling. In addition, the decreased synaptic protein levels observed in the muscle following neuronal Wnd signaling may be result from modulation of Tor activity, as postsynaptic overexpression of Tor globally elevates protein synthesis[41] and partially restores receptor levels (Fig. 6). Finally, an intriguing study in the rodent visual system revealed that ablation of presynaptic photoreceptors leads to remodeling of the postsynaptic apparatus, including the rapid and localized disappearance of GluRs[22]. Together, these studies demonstrate that postsynaptic targets adapt to injury, disease, and loss-of-presynaptic inputs by remodeling postsynaptic neurotransmitter receptor complexes to reduce sensitivity, suggesting a conserved response.

One mechanism that stabilizes the reduced state of synaptic strength following neuronal Wnd signaling is a selective occlusion of postsynaptic PHP transduction. Neurons experiencing Wnd signaling are competent to homeostatically modulate presynaptic neurotransmitter release, but apparently do not receive the retrograde information necessary from the muscle, even following additional perturbations to or restorations of postsynaptic GluRs. Interestingly, perturbations to Cap-dependent protein synthesis in the postsynaptic muscle have been shown to disrupt PHP retrograde signaling[42], and metabolic changes in the muscle can also impinge on this pathway to modulate PHP signaling[51]. This indicates that one mechanism utilized by the muscle to respond to neuronal Wnd signaling, distinct from a general reduction in protein synthesis, may be a selective inhibition of Tor-dependent protein synthesis which, in turn, contributes to the occlusion of PHP signaling. It is interesting to note that while PHP signaling is highly compartmentalized[52], it can be still be expressed in neurons despite perturbations to a variety of processes that disrupt synaptic structure and function[37,53,54] independently of injury-related signaling. Thus, an instructive cue mediated by Wnd signaling selectively impairs retrograde PHP communication in the muscle to stabilize a reduced level of synaptic strength.

Although PHP signaling appears to be inhibited, a second mechanism stabilizes the reduced set point of synaptic strength following neuronal Wnd signaling, GluR scaling. This postsynaptic form of homeostatic plasticity parallels the postsynaptic scaling of GluRs observed following silencing of neuronal activity in mammalian central synapses[55]. At the *Drosophila* NMJ, the induction of this form of homeostatic plasticity was known to require hypo-innervation[30], and we have shown that a selective increase in postsynaptic GluRIIA-containing receptors compensates for reduced neurotransmitter release and maintains stable synaptic strength. Although the signal transduction system that mediates this form of plasticity is enigmatic, it is clearly distinct from the retrograde signaling system that drives PHP. The homeostatic set point of synaptic strength has been demonstrated to be plastic, and can be stabilized at levels distinct from baseline values in mutations that disrupt synaptic function[37,45,53,54] and during aging of the synapse[56]. Our results define injury-related signaling as an additional process that has the capacity to adjust the homeostatic set point of synaptic strength.

Why might synaptic strength be adjusted to a reduced level following injury? The coordinated reduction in Wnd-mediated postsynaptic responsiveness and Hiw-mediated presynaptic efficacy may promote sufficient time for the process of repair vs. degeneration to be adjudicated within an injured neuron, while still maintaining synaptic communication at a reduced level. Loss or further reductions to neurotransmission may destabilize synaptic integrity, impairing the series of subsequent steps necessary to restore normal synaptic strength and structure should the injury be successfully overcome. In addition, a further reduction in postsynaptic receptivity may deprive the neuron of necessary trophic support and promote neuronal degeneration. Indeed, a lack of trophic support from the muscle due to weakened synaptic activity contributes to neuromuscular disease pathogenesis[57]. In the central nervous system, there is evidence that synaptically coupled cells sense and respond to injury[20,21,23]. Hence, injury to an individual neuron can propagate and destabilize an entire neural circuit without adaptive counter measures. Our findings illustrate the acclimations that occur in postsynaptic targets to neurons experiencing injury-related signaling and highlight the adaptations to synaptic plasticity that maintain stable functionality around a reduced set point of synaptic strength.

## Methods

**Drosophila stocks.** *Drosophila* stocks were raised at 25 °C on standard molasses food. The $w^{1118}$ strain is used as the wild-type control unless otherwise noted, as this is the genetic background of the transgenic lines and other genotypes used in this study. The following fly stocks were used: $hiw^{\Delta N33}$, $wnd^3$ and $wnd^{dfED22810}$, $GluRIIA^{SP1638}$, UAS-wnd[10], OK319-Gal4[58], c380-Gal4[59], OK6-Gal4[60], elav-GeneSwitch[35]; MHC-Gal4[61], UAS-TNT[36], UAS-Tor[62], puc-lacZ[E699]; UAS-dGluRIIA[63], UAS-FasIIA-PEST+[58]. Biased innervation was achieved using the following genotype: tubP>stop>Gal4, UAS-FLP, UAS-CD8-GFP/UAS-FasIIA-PEST+; H94-Gal4, nSyb-Gal80/+, as described[64]. All other stocks were obtained from the Bloomington Drosophila Stock Center. Standard chromosomal balancers and genetic strategies were used for all crosses and for maintaining mutant lines.

**GeneSwitch experiments.** Acute activation of Wnd signaling was achieved through administration of RU486 or saline control to larvae (elav-GeneSwitch; UAS-wnd) as described[35]. Briefly, early third-instar larvae were raised on standard molasses media and then food containing saline or RU486 (25 µg/mL; Mifepristone; Sigma, St. Louis, MO) for 24, 48, or 72 h and then dissected for immunostaining or electrophysiology.

**Immunocytochemistry.** Third-instar larvae were dissected in ice cold 0 $Ca^{2+}$ HL-3 and fixed in either Bouin's fixative for 2 min or 4% paraformaldehyde for 10 min as described[65]. Larvae were washed with PBS containing 0.1% Triton X-100 (PBST) for 30 min, blocked for 1 h in 5% Normal Donkey Serum (NDS), overnight incubation in primary antibodies at 4 °C, then washed in PBST, incubated in secondary antibodies for 2 h, a final wash in PBST, and equilibration in 70% glycerol. Samples were mounted in VectaShield (Vector Laboratories). The following antibodies were used: mouse anti-Bruchpilot (BRP; nc82; 1:100; Developmental Studies Hybridoma Bank, DSHB); mouse anti-DLG (4F3; 1:100; DSHB), mouse anti-GluRIIA (8B4D2; 1:100; DSHB); rabbit anti-GluRIIB (1:1000; generated by Cocalico Biologicals using the peptide describe in ref. [33]); rabbit anti-GluRIII (1:2000; generated and affinity purified by Cocalico Biologicals using the peptide described in ref. [66]); guinea pig anti-vGlut (1:1000; generated by Cocalico Biologicals using the peptide described in ref. [67]; guinea pig anti-GluRIID (1:1000; generated by Cocalico Biologicals using the peptide described in ref. [66]). The following secondary antibodies were obtained from Jackson ImmunoResearch: Alexa Fluor 647 conjugated goat anti-HRP (1:200; #123-605-021), Alexa Fluor 488 conjugated donkey anti-mouse (1:400; #715-545-150), Alexa Fluor 405 conjugated donkey anti-guinea pig (1:400; #706-475-148), Alexa Fluor Cy3 conjugated donkey anti-guinea pig (1:400; #706-165-148), and Alexa Fluor Cy3 conjugated donkey anti-rabbit (1:400; #711-165-152).

**Imaging and analysis.** Samples were imaged using a Nikon A1R Resonant Scanning Confocal microscope equipped with NIS Elements software and a ×100 APO 1.4NA oil immersion objective using separate channels with four laser lines (405, 488, 561, and 647 nm). For fluorescence intensity quantifications of vGlut, Dlg, GluRIIA, GluRIIB, and GluRIID, z-stacks were obtained using identical gain and laser power settings with z-axis spacing between 0.15 and 0.2 µm for all genotypes within an individual experiment. Maximum intensity projections were utilized for quantitative image analysis using the general analysis toolkit of NIS Elements software. The fluorescence intensity levels of vGlut, Dlg, GluRIIA, GluRIIB, and GluRIID immunostaining were quantified by applying intensity thresholds and filters to binary layers in the 405, 488, and 561 nm labeled channels.

The mean intensity for each channel was quantified by obtaining the average of the total fluorescence signal of each individual vGlut, Dlg, or GluR puncta and dividing this value by the area of each specific puncta. The sum intensity for GluRIIA, GluRIIB, and GluRIID was quantified as the total fluorescence signal of each individual GluR puncta. A mask was created around the HRP channel, used to define the neuronal membrane, and only puncta within this mask were analyzed to eliminate background signal. To quantify receptor density, a mask of Dlg over the entire NMJ was used to determine the total postsynaptic area and total receptor number at that particular NMJ was then divided by this area. To assay bouton numbers, both Type Ib and Is boutons were counted using vGlut and HRP-stained NMJ terminals on muscle 4 of segment A2, considering each vGlut puncta to be a bouton. All measurements based on confocal images were taken from synapses acquired from at least six different animals. The mean intensity of puc-lacZ expression was quantified as described[9], in which neuronal nuclei along the dorsal midline of the nerve cord in segments A3–A7 were imaged. Mean lac-Z intensity was measured for at least five animals of each genotype.

**Electrophysiology**. All dissections and recordings were performed in modified HL-3 saline[54,68,69] containing (in mM): 70 NaCl, 5 KCl, 10 MgCl$_2$, 10 NaHCO$_3$, 115 sucrose, 5 trehalose, 5 HEPES, and 0.3 CaCl$_2$, pH 7.2. Neuromuscular junction sharp electrode (electrode resistance between 10 and 35 MΩ) recordings were performed on muscles 6 and 7 of abdominal segments A2 and A3 in wandering third-instar larvae. Larvae were dissected and loosely pinned; the guts, trachea, and ventral nerve cord were removed from the larval body walls with the motor nerve cut, and the preparation was perfused several times with HL-3 saline. Recordings were performed on an Olympus BX61 WI microscope using a ×40/0.80 water-dipping objective. Recordings were acquired using an Axoclamp 900 A amplifier, Digidata 1440 A acquisition system and pClamp 10.5 software (Molecular Devices). Electrophysiological sweeps were digitized at 10 kHz, and filtered at 1 kHz. Data were analyzed using Clampfit (Molecular devices), MiniAnalysis (Synaptosoft), Excel (Microsoft), and SigmaPlot (Systat) software.

Miniature excitatory postsynaptic potentials (mEPSPs) were recorded in the absence of any stimulation, and cut motor axons were stimulated to elicit excitatory postsynaptic potentials (EPSPs). Noise analysis (±0.03 mV) demonstrated that small mEPSP amplitudes could be reliably detected above background in all genotypes and conditions (Supplementary Table 1). An ISO-Flex stimulus isolator (A.M.P.I.) was used to modulate the amplitude of stimulatory currents. Intensity was adjusted for each cell, set to consistently elicit responses from both neurons innervating the muscle segment, but avoiding overstimulation. For each recording, at least 100 mEPSPs were analyzed to obtain a mean mEPSP amplitude value. The average single-AP-evoked EPSP amplitude of each recording is based on 20 EPSPs. Quantal content was estimated for each recording by calculating the ratio of mean EPSP amplitude to mean mEPSP amplitude and then averaging recordings across all NMJs for a given genotype. Muscle input resistance ($R_{in}$) and resting membrane potential ($V_{rest}$) were monitored during each experiment. Recordings were rejected if the $V_{rest}$ was above −60 mV, if the $R_{in}$ was <5 MΩ, or if either measurement deviated by >10% during the course of the experiment. Larvae were incubated with or without philanthotoxin-433 (Sigma; 20 µM) resuspended in HL-3 for 10 min as described[37].

**Statistical analysis**. Statistical analysis was performed using GraphPad Prism software. Data were tested for normality using a D'Agostino-Pearson omnibus normality test. Normally distributed data were analyzed for statistical significance using a t-test (pairwise comparison), or an analysis of variance (ANOVA) and Tukey's test for multiple comparisons. For non-normally distributed data, Wilcoxon rank-sum test or Dunn's multiple comparisons after nonparametric ANOVA were used. All data are presented as mean ± SEM with varying levels of significance assessed as *$p < 0.05$, **$p < 0.01$, ***$p < 0.001$, ****$p < 0.0001$, NS not significant. See Supplementary Table 1 for additional statistical details and values.

**Data availability**. The data that support the findings of this study are available from DD upon reasonable request. The authors declare that the data supporting the findings of this study are available within the paper and its Supplementary Information files.

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

## Acknowledgements

We thank Catherine Collins (University of Michigan, USA), Pejmun Haghighi (Buck Institute, USA), Brian McCabe (Ecole Polytechnique Federale de Lausanne, Switzerland), and Aaron DiAntonio (Washington University, USA) for sharing *Drosophila* stocks. We also thank Catherine Collins for insightful comments and discussions. We acknowledge Andrew Frank, Tim Mosca, and Sarah Perry for technical help and advice on RU486 experiments. We also acknowledge the Developmental Studies Hybridoma Bank (Iowa, USA) for antibodies and the Bloomington Drosophila Stock Center (NIH P40OD018537) for stocks used in this study. P.G. was supported in part by a USC Provost Graduate Research Fellowship. This work was supported by a grant from the National Institutes of Health (NS091546) and research fellowships from the Alfred P. Sloan, Ellison Medical, Whitehall, Mallinckrodt, and Klingenstein-Simons Foundations to D.D.

## Author contributions

P.G. performed all experiments and analyzed all data. P.G. and D.D. wrote the manuscript.

## Additional information

**Competing interests:** The authors declare no competing interests.

