## [Peer Review File · Nature Communications]

Reviewers' comments:

Reviewer #1 (Remarks to the Author):

This study by Goel and Dickman explored the injury-signaling DLK pathway at the *Drosophila* NMJ and its interaction with the known presynaptic homeostatic plasticity (PHP) mechanisms under uninjured conditions. They found that activation of DLK signaling, through either deletion of a E3 ubiquitin-ligase highwire (*hiw*) or overexpression of a *hiw* substrate Wallenda (*Wnd*), turns on the degenerative signaling cascade by suppressing PHP through inhibiting a postsynaptic mechanism that is normally engaged by postsynaptic muscle cells to retrogradely signal presynaptic neuronal terminal for activation of PHP. Indeed, when inhibition on retrograde signaling is bypassed by overexpression of Tor in the muscle cells, which presumably acts downstream of the point of *Wnd*/DLK suppression, PHP is restored. Thus, results from this study discovered a decision-making mechanism that allows a switch between PHP in healthy intact NMJ and initiation of degenerative process in injured NMJ, including engaging a separate postsynaptic homeostatic mechanism to stabilize the diminished state of neurotransmission.

Overall, this is a very interesting study with many elegantly designed experiments, and is potentially suitable for publication in *Nature Communication*. However, there are a few conceptual gaps that needs to be addressed/discussed.

Figure 5 claims that *hiw*-mutant NMJs can express PHP when Tor is overexpression postsynaptically. However, what the results showed is that in postsynaptic Tor-OE leads to increased quantal content in *hiw*-mutants, which is similar to WT with Tor-OE. The authors cannot exclude the possibility that Tor-OE does something entirely different than PHP by simply promote pre- and post-synaptic functions in the *hiw*-mutants (evidenced by the increase in both presynaptic release and postsynaptic GluR expression). PhTx- or GluIIA mutant should be used to demonstrate acute or chronic PHP is indeed restored.

In both *hiw*-mutant and *Wnd*-OE, postsynaptic GluR expression is greatly reduced. Hypo-innervation induced by FasII-OE at muscle 6 of *hiw*-mutant triggers a postsynaptic homeostatic synaptic plasticity mechanism that increased GluR expression. What exactly is the trigger of the postsynaptic homeostatic mechanisms? How does the muscle cell distinguish hypo-innervation from *hiw*-deletion as both leads to reduced quantal content? Is the hypo-innervation induced postsynaptic homeostatic plasticity operational in the *hiw*;*wnd* double mutant?

Data from Supplementary figures show that both *hiw* and *wnd* work presynaptically to suppress postsynaptic induction of retrograde PHP mechanism, and that evoked synaptic transmission is not required for such pre- to post- communication. Can the authors speculate then what may be a candidate mechanism that is independent of tetanus toxin-sensitive vesicular release? Is it possible that this is through a direct diffusion mechanism or interactions of trans-synaptic adhesion molecules?

Reviewer #2 (Remarks to the Author):

In this manuscript, Goel and Dickman investigate structural and functional adaptations at synapses of the *Drosophila* neuromuscular junction upon activation of signaling pathways that mediate responses to neuronal injury. The authors characterize to what degree homeostatic plasticity mechanisms operate under these conditions. Specifically, they provide a very thorough investigation of regulatory functions of the DLK *Drosophila* Homolog Wallenda (Wnd) and the PHR protein Highwire (hiw). Consistent with previous results, the authors conclude that both proteins exert their regulatory function at the presynapse to affect postsynaptic glutamate receptors. Highwire is a negative regulator of Wnd and the deactivation of hiw causes an increase of presynaptic Wnd levels to induce a trans-synaptic signaling that affects the postsynaptic Glutamate receptors. In agreement with previous observations by Collins et al. (2006), an additional function of hiw in presynaptic vesicle release is revealed in hiw;Wnd double mutants. The authors here report that Highwire also influences presynaptic release by affecting the number of vesicles released per action potential (quantal content). Following Highwire loss, the susceptibility of the synapse to two forms of homeostatic plasticity was investigated and the authors report that a plasticity mechanism that resets transmission strength following connection loss is maintained. In contrast, the authors conclude that another plasticity mechanism, presynaptic homeostatic plasticity, that operates to increase neurotransmitter release and is induced by the acute or chronic reduction of postsynaptic glutamate sensitivity, is lost. The authors further conclude that this loss is due to an integration step upstream of the retrograde trans-synaptic (post- to presynapse) signaling, as direct activation by induction of postsynaptic Tor signaling is still capable of inducing presynaptic homeostatic plasticity. Especially the comparative analysis of structural and functional adaptations and the rescue mechanisms that exist to maintain or degenerate synaptic connections is a timely and highly relevant matter. The study is thus of wide-spread interest and I recommend its publication upon appropriate revision. The manuscript is well-written and the experimental data are sound. However, some important aspects (detailed below) need to be clarified prior to publication.

Major points:

1. In this study the effects of Hiw and Wnd signaling on structural and functional adaptations of synapses is thoroughly described. However, the link to the axonal injury that induces Wnd signaling is less clear. In my eyes this is not a problem per se, but I suggest shortening the relatively long sections in the introduction & discussion dedicated to this. The additional space could be used to discuss the relation between the morphological and functional changes (including the role of DLG) induced by Wnd signaling/hiw loss in greater detail. Furthermore, a more detailed discussion/speculation of the additional role of Hiw on presynaptic release (independently of Wnd signaling, revealed in the double mutant) would be of great interest.
2. It is necessary to report on the mEPSC event frequencies in all experiments.
3. In cases in which normalized data are shown, the non-normalized absolute values need to be included in table 1.
4. Loss of hiw results in a decreased size of mEPSCs (Fig. 1B). In addition, postsynaptic glutamate receptor fields are clearly altered (Fig. 1E). Yet it is not entirely clear whether

these alterations fully account for the reduced mEPSC amplitudes observed. From the images depicted in Fig. 1E it looks that the intensity per spot is not affected as much as the density/number of particles. In fact, the fluorescence intensity per cluster looks rather similar between groups. Supposing that each single spot corresponds to one synaptic connection, it is difficult to understand how this could have such a strong effect on the mEPSC size. This is contrasted by a ~90% reduction in GluRIID levels as implied by the quantification in Fig 1H, which does not match the images. Please include a clearer description of how the fluorescence images are quantified in the methods section (measured per spot or intensity measured over the whole NMJ?). The authors need to quantify and report the spot-intensity, their number and density per NMJ. An image depicting the full junction (similar to Wan et al., 2000; Neuron) would be helpful as a supplemental item. It is further necessary to see the quantification of some presynaptic marker (e.g. BRP) in all conditions depicted here (from the text in the methods section it is evident that BRP stainings were already performed, so please include these). The number of presynaptic active zones should be quantified and co-staining of BRP and GluRIID investigated to establish whether active zones and glutamate receptors remain strictly opposed in the conditions studied. Furthermore, it would be interesting to relate the levels of vGlut to the amounts of BRP to support the notion that the postsynaptic degeneration is the dominating influence for the reduction of the quantal size.

5. An unresolved issue is whether the postsynaptic morphological changes of the glutamate receptor fields (particularly their number/density) are cause or consequence of the lower mEPSC size. Could the authors attempt to restore the mEPSC amplitude (by e.g. presynaptic over-expressing vGlut) and test, whether this influences the morphology? It may be difficult to achieve this experiment in the short timeframe of a revision in hiw mutants, but alternatively combined Wnd and vGlut overexpression could be performed.

6. The interpretation of whether the quantal content is reduced upon Wnd overexpression is difficult. It looks like there is a clear trend towards reduction compared to wildtype flies (p value of 0.1). The authors should repeat the experiments with a comparison to animals expressing the Gal4 driver, which may give a clearer result.

7. The effect observed on DLG (Figs. S2 and 2) is very interesting, but its relation to the effects of Wnd signaling on the glutamate receptors and on synaptic transmission should be further investigated. For instance, does postsynaptic DLG overexpression revert the effects caused by either hiw loss or presynaptic Wnd overexpression?

8. The morphological changes observed upon GluRIIA overexpression (Fig. 6A) are very interesting. Is the density/number of GluRIIA, -B,-D puncta significantly increased? Could the authors please investigate whether DLG is affected by GluRIIA OE in hiw mutants (Figs. S2&2)? It is interesting that in the condition of hiw the usual competition/between GluRIIA and -B seems to be gone upon loss of hiw. Did the authors test whether overexpression of GluRIIB would induce a similar effect on GluRIIA levels and whether this preference depends on DLG? Experiments could be done relatively easily by combining an overexpression of GluRIIA or -B with Wnd.

9. The interpretation that hiw loss reduces the quantal content relies heavily on the adequate detection of mEPSCs in this genotype. This is cumbersome, as these must be close to the detection limit. An over-estimation of the mEPSC amplitude would lead to an under-estimation of the quantal content and a smaller quantal content is observed in this genotype (Figs. 1, 2, S2). This problem becomes worse when mEPSC amplitudes are further

reduced upon application of PhTx or GluRIIA mutation. Both treatments further reduce the quantal content (Figs. 3, 4), which is worrisome. Therefore, it is difficult to be certain that the homeostatic plasticity is fully blocked, or whether it is present, yet not observable because mEPSCs are reduced below the detection limit. Indeed, a cutoff of mEPSCs below certain, observable amplitudes may explain the unusually small variance in this group (revealed by the unusually small error bars, Fig.3B, 4D). In my opinion the experiments that combine PhTx treatment with GluRIIA overexpression (Fig. 6) do not alleviate this concern, because these reveal a clearly higher quantal content in hiw mutants following administration of PhTx, adding to this concern (in Fig. 6G this close to 100% compared to 50% in Fig 3). Thus, an alternative interpretation of the data could be that hiw mutants are still capable of PHP, which remains undetected due to small mEPSCs. The authors should at least discuss this or tackle this concern experimentally by manipulations that increase mEPSC sizes to the same levels as controls in hiw mutants (e.g. *v*-Glut overexpression, altered saline solution).

10. The experiments in which mTor signaling is activated nicely demonstrate that PHP can be induced in hiw mutants. But as this treatment also increases the mEPSC size, the above concern regarding the detection threshold pertains. An intriguing aspect of these data is that also here the number/density of glutamate receptor fields appears to increase (Fig. 6A). The authors should quantify this. Could the authors further investigate this condition by checking whether presynaptic Wnd levels are affected (an antibody was described by Collins et al., 2006)?

Minor points:

11. Figure S2: Include data from panels A/B in table 1.

12. Figure 2: What is the effect of TNT overexpression on the wildtype DLG and GluR levels?

13. Figure 5G looks like it is mislabeled, how can the increase in GluRIID be smaller than for GluRIIA and -B?

14. Figure 6G: Check labelling, correct data & color code (it is unclear how the condition hiw +GluRIIA-OE + PhTx can deviate significantly from 100% yet have a similar mean). Please explain how normalization and statistical testing was performed. Include non-normalized data in Table 1 here and all other instances where normalization was performed.

15. Line 60: please spell out what "PHR" stands for upon first use

16. Line 137: add "directly" to the following sentence: "but does not DIRECTLY affect mEPSP amplitude" to qualify that this effect is likely due to its regulation of Wnd.

17. Line 175: please clarify that AP-EVOKED "synaptic activity" is meant.

18. Line 180: please clarify this statement, as it is unclear what is meant by the "postsynaptic calcium response". Which of the experiments in Newman et al., does this refer to? Is this the Ca²⁺ influx per AP-evoked event? Or the number of events by an action potential? Include "during AP-EVOKED neurotransmission" and clarify this statement.

19. Figure 8H right panel: This drawing is confusing, because the black arrow is crossed, but the retrograde signaling is intact upon Tor signaling.

20. Line 544: which experiments are meant with "otherwise specified"?

Altogether this is a thorough investigation of structural and functional adaptation of synapses upon hiw loss/ induction of Wnd signaling. The manuscript is well written, the data of high quality and the findings are of high interest to the neuroscience community. In the

current form, the manuscript is strongly focused on the description of the effects of hiw loss on quantal content. While this is a valid interpretation in the light of the data, the exact measure is experimentally difficult to obtain. Therefore this must either be validated or alternative scenarios discussed. To extend the scope of this investigation, some additional experiments focusing on the causal relation between mEPSC-size, the morphology of postsynaptic glutamate receptor fields and DLG should be attempted during a revision.

Reviewer #3 (Remarks to the Author):

In this paper the Dickman group investigates how Wnd signaling from the presynaptic terminal affects responses in the post synaptic muscle at *Drosophila* larval neuromuscular junctions. DLK is known to increase when axons are injured, and thus, over expression of DLK is used here as a model for axonal injury. The authors find that increased DLK causes the NMJ to overgrow and boutons to undergrow. In the post synaptic muscle, glutamate receptors and DLG/PSD95 are reduced. While the characterization thus far is standard, in the next part the strength of the Dickman group surfaces: Lower GluR abundance normally triggers a compensatory homeostatic regulatory mechanisms that ensures steady presynaptic transmitter release, but in these mutants with too much DLK this apparently does not happen. What is exciting about their finding is that the animals with too much DLK have not lost the ability to upregulate presynaptic release all together, because upregulation of Tor in muscles is still able to increase presynaptic release. Hence, the authors conclude there are different mechanisms of homeostatic regulation at *Drosophila* NMJs and this is important during recovery from axonal injury. The paper also concludes that the decreased neurotransmitter set point they observe upon DLK over expression may be a protective mechanism to the injured synapse. While this latter point was not directly shown, it is an interesting idea to think about.

This is a well written manuscript with great potential and an interesting, maybe even provocative message. The flow is good, but I am a little bit concerned about overall interpretation as outlined below:

- 1) The paper is light on mechanism. What are the signaling molecules, how is DLK affecting the post synaptic cell and how is the presynaptic terminal strength in turn set by this novel pathway?
- 2) The NMJs appear morphologically quite disrupted and I am not sure if the homeostatic phenotypes described occur independently from these defects. What I mean is that over expression of DLK is used as a proxy for axonal injury but this protein is always expressed and unlike what happens in 'real' injury, I imagine that in the case of the authors, the synapse never properly developed. It would be interesting to see if in a real injury model (where DLK also becomes upregulated following injury) similar observations are made as compared to the constitutive expression of DLK. In parallel, acute induction of DLK expression could be attempted as well.

RESPONSE TO REVIEWERS

We thank the Reviewers for their constructive comments and for judging our manuscript to be of significant interest and high quality. We appreciate the favorable opinions by all three Reviewers, finding the manuscript to be potentially suitable for publication given that attention is paid towards addressing their valid concerns and suggestions for improvement. The major comments were to perform additional controls to determine if postsynaptic Tor overexpression has any impact on presynaptic DLK signaling, to perform additional analyses and descriptions of postsynaptic GluRs, and to attempt to uncouple the impacts of chronic DLK signaling on presynaptic structure and postsynaptic remodeling. In response, we have addressed all of these concerns with new experimental data, additional analyses, and significant improvements to the text. This includes one new Figure and four additional Supplementary Figures in addition to significant textual changes to the revised manuscript. Together, these revisions have substantially improved the manuscript. We hope the Reviewers and the Editor agree that this manuscript is now suitable for publication at *Nature Communications*.

Response to Reviewer 1

1. Reviewer 1 is concerned that the results shown in Fig. 5, in which PHP is restored in *hiw+Tor-OE*, may be due to a non-specific influence of Tor-OE on *hiw* synapses. This Reviewer suggests testing whether PHP is actually restored in this condition by applying PhTx and assaying quantal content.

We thank this Reviewer for raising this valid point and agree this is an important control to include. First, we should mention that during the time this manuscript was in review, our lab published a paper showing that PhTx application, *GluRIIA* mutants, and Tor-OE all converge to induce the same retrograde signaling system that induces PHP expression (Goel et al., 2017). In that paper, we showed that PhTx application to a Tor-OE synapse (in an otherwise wild-type background) reduces mEPSP amplitude (as expected) but also reduces EPSP amplitude, with no change in quantal content (Figure 1 of the *Cell Reports* paper). We interpreted this result to be due to Tor-OE having already fully induced PHP expression, so PhTx application could not induce the same PHP signaling system over again. Thus, no change in quantal content was observed, and a reduction in EPSP amplitude was observed.

Therefore, in performing the experiment that Reviewer 1 astutely suggested, we would expect the mEPSP amplitude should be reduced with PhTx application to *hiw+Tor-OE* NMJs. If PHP were induced and expressed in *hiw+Tor-OE* to begin with, as we suggest, then PhTx application should not change quantal content, and EPSP amplitude should be reduced. Indeed, this is exactly what we observed. These new results are now shown in **Figure 5A-E** and discussed on **Page 12 Lines 376-378** in the revised manuscript. We thank the Reviewer for suggesting this important control.

2. Reviewer 1 questions how hypo-innervation on muscle 7 induces the homeostatic scaling of postsynaptic GluR levels shown in Figures 7 and 8, and in particular what the induction mechanism is for this form of homeostatic plasticity?

The Reviewer brings up many fascinating and important questions about the induction mechanisms and signaling system that drives homeostatic receptor scaling in response to hypo-innervation. The manipulation and electrophysiological adaptations that involved in this novel form of homeostatic plasticity were initially described in the seminal study cited in our manuscript (Davis and Goodman, 1998). In this current study, we are the first to define the

expression mechanism of this form of plasticity, namely a homeostatic enhancement in the levels of postsynaptic GluRIIA-containing receptors (Figure 7). Further, we demonstrate that the postsynaptic signaling system that orchestrates this form of adaptive plasticity is distinct from that which underlies retrograde PHP signaling. We believe these are important and novel insights that will motivate and inform additional studies into hypo-innervation-induced receptor scaling.

We absolutely agree with the Reviewer that fundamental questions about nature of this homeostat – the induction mechanism, whether reduced glutamate release, trans-synaptic adhesion complexes, calcium signaling, and the postsynaptic signal transduction system that ultimately leads to enhanced GluRIIA-containing receptor expression – are important and need further study.

[redacted]

However, the current manuscript is focused on the impact of presynaptic Wnd/DLK-signaling on postsynaptic receptors, synaptic strength, and homeostatic plasticity pathways. We therefore feel that the fascinating questions posed by the Reviewer, which are of significant interest, are outside the scope of the current manuscript.

[redacted]

3. Reviewer 1 wonders whether hypo-innervation induced postsynaptic receptor scaling is operational in the *hiw;wnd* double mutant?

We appreciate this question. We show in the current manuscript that postsynaptic homeostatic receptor scaling is intact and fully expressed in *hiw* mutants and in otherwise wild-type synapses. We have attempted to perform the experiment suggested by the Reviewer, but unfortunately could not generate the complicated genetic lines necessary, despite attempting several different strategies. In particular, *hiw* mutants are unhealthy to begin with, and we have not been able to generate a viable stock that includes the H94-Gal4/UAS-fasII chromosomes in combination with the *wnd* mutant allele. However, the *hiw;wnd* double mutant effectively rescues most aspects of NMJ morphology, growth, and postsynaptic functionality. So we have no reason to believe that postsynaptic receptor scaling would not be expressed properly in the *hiw;wnd* double mutant, as it is in control and *hiw* mutants alone.

4. Reviewer 1 suggests that we speculate about a possible mechanism to explain how presynaptic *wnd* signaling is communicated to the postsynaptic target in the absence of evoked synaptic vesicle fusion.

This is indeed a fascinating question and one we agree deserves more discussion. Trans-synaptic “nanocolumns” have emerged as inter-cellular signaling complexes that orchestrate many aspects of synaptic development, structural alignment, and signaling during maturation of synapses and during various forms of plasticity (see timely review (Biederer et al., 2017)). In this review, trans-synaptic cell adhesion molecules were proposed to mediate signaling through nanocolumns between pre- and post-synaptic compartments, a mechanism that does not require neurotransmitter release. An attractive possibility, therefore, is that synaptic signaling through these trans-synaptic nanocolumns, perhaps mediated through inter-cellular adhesion complexes, may communicate injury-related signaling from the presynaptic terminal to the postsynaptic partner. This, in turn, may lead to destabilization of synaptic contacts to alter synaptic structure, postsynaptic scaffolds, and postsynaptic GluRs. Thus, as pointed out by the Reviewer, cell adhesion molecules may mediate activity-independent signaling during activation

of the Wnd pathway. We now discuss this possibility in the Discussion of the revised manuscript (**Page 19 Lines 560-571**) and thank the Reviewer for encouraging us to more fully consider this intriguing question.

Response to Reviewer 2

1. Reviewer 2 suggests we provide a more in depth discussion regarding the morphological and functional changes induced by Wnd signaling, as well as some speculation about the additional role of Hiw on presynaptic release.

We thank this Reviewer for urging us to more fully consider these topics and we absolutely agree. As suggested by the Reviewer, we have shortened some areas in the Introduction and Discussion regarding axonal injury and Wnd signaling, and have provided more discussion about the morphological and functional consequences of Wnd signaling. Indeed, we now have included an entirely new Figure (Figure 2) devoted to uncoupling the structural and functional changes at NMJs induced by Wnd signaling (see Response to Reviewer 3 below). Further, we now add additional discussion regarding the Wnd-independent functions of Hiw in reducing presynaptic release, which we now highlight as a mechanism that may enable functional adaptations on both sides of the synapse during injury-related signaling. These changes are now included on **Page 18 Lines 536-546** of the revised manuscript.

2. Reviewer 2 requests mEPSP event frequencies be reported for all experiments.

We agree that mEPSP frequencies should have been included in the original manuscript. We have now detailed all mEPSP frequencies for each data set in the revised **Table S1**.

3. Reviewer 2 requests that all absolute values for normalized data be included in Table S1.

This data was included in the original manuscript, along with additional statistical data corresponding to each Figure. We have double checked and made sure that all previous and new data are included in the revised manuscript and have ensured that all absolute values for normalized data are explicitly shown in the revised **Table S1**.

4. Reviewer 2 requests that we include additional analyses and statistics regarding the characterization of postsynaptic GluRs in hiw and wnd-OE synapses. Further, he/she requests additional data regarding the number of active zones, receptor clusters, and synaptic vesicle markers.

We thank the Reviewer for these suggestions and agree that additional experimental data and analyses will improve our characterization of synapse structure in this manuscript. In the revised manuscript, we have now included additional analyses of GluR levels, including GluR puncta size, intensity, puncta number per NMJ, and density now presented in a new **Supplemental Figure S2**. We further discuss these results in the revised manuscript, showing that the total number and density of all GluR subunits are reduced at *hiw* and *wnd*-OE NMJs. Further, we have now included detailed information about the size and fluorescence intensity (both mean and sum) of GluR puncta, with further details included in the revised Methods section. Finally, as suggested by the Reviewer, we have now included entire NMJ images in the new Figure S2 to highlight the overall reduction in GluR intensity levels in *hiw* and *wnd*-OE. These insights are now discussed on **Page 7 Lines 205-211**.

In addition, as suggested by the Reviewer, we have provided a new supplemental figure in the revised manuscript (**Figure S1**). In this figure, we now show BRP, GluRIIC, vGlut, and HRP staining at an entire NMJ synapse. We also quantify neuronal membrane area, vGlut intensity, BRP puncta density, and the apposition of BRP-GluRIIC puncta. The results of this analysis are now discussed in the Results section (**Page 6 Lines 176-180**). We thank the Reviewer for suggesting these experiments and additional analyses, which have improved the manuscript.

5. Reviewer 2 suggests we attempt to restore mEPSP amplitude in *hiw* mutants or *wnd*-OE by presynaptic overexpression of vGlut.

As the Reviewer notes, presynaptic overexpression of the vesicular glutamate transporter leads to enlarged synaptic vesicle size, increased glutamate released from each vesicle, and enhanced mEPSP amplitude (Daniels et al., 2004). This is a great experiment that, in fact, we had attempted while preparing the original manuscript. Unfortunately, expression of vGlut in a *hiw* mutant background failed to change mEPSP amplitude (see Reviewer Figure 1), indicating vGlut-OE failed to have the intended impact.

[redacted]

[redacted]

6. Reviewer 2 requests comparing quantal content in *wnd*-OE to quantal content from a control genotype consisting of the *Gal4* driver alone.

We appreciate this point and agree that it is important to clearly establish whether quantal content in *wnd*-OE is significantly different from wild-type synapses. As requested by the Reviewer, we have now recorded from *c380-Gal4/+* animals and performed additional recordings in *wnd*-OE. This analysis has revealed no significant difference in mEPSP amplitude, EPSP amplitude, nor quantal content in *c380-Gal4/+* synapses compared to wild type, while mEPSP and EPSP amplitudes in *wnd*-OE are reduced. Importantly, quantal content is unchanged in *wnd*-OE compared to both wild-type and *c380-Gal4/+* animals. This data is now shown in the revised **Table S1**, mentioned in the **Figure 1** legend, and discussed in the text on **Page 6 Lines 184-185**. We thank the Reviewer for urging us to include this additional control.

7. Reviewer 2 finds the changes shown in Fig. 2 and S2 detailing postsynaptic DLG levels in *hiw* mutants interesting, and wonders whether muscle overexpression of DLG can compensate for the impacts of presynaptic *wnd* signaling?

We agree that the change in postsynaptic scaffolding induced by presynaptic *Wnd* signaling is very interesting and worthy of further study.

[redacted]

That being said, we have attempted to perform the experiment requested by the Reviewer, namely overexpressing DLG in the postsynaptic muscle of *hiw* mutants. Unfortunately, we were unable to generate any viable larvae of the correct genotype (*hiw; BG57-Gal4/UAS-DLG*). This may be due to incompatible genetic backgrounds, given that *hiw* mutants are very sick to begin with, or perhaps some toxicity introduced by high expression of DLG in *hiw*-mutant muscle.

[redacted]

8. Reviewer 2 requests additional analyses of GluRs and DLG levels in *hiw*+GluRIIA-OE, and questions what happens if GluRIIB is overexpressed in *hiw* mutants?

We thank the Reviewer for these questions and we agree it would be interesting to determine what impact GluRIIB-OE may have on *hiw*-mutant muscle. First, we have included additional analyses of GluRs in *hiw*+GluRIIA-OE, as suggested by the Reviewer. We find that in addition to increased intensity levels of individual GluR puncta (quantified in **Figure 7B**), the number and density of GluRIIA and GluRIID receptor puncta per NMJ are also increased in *hiw*+GluRIIA-OE compared to *hiw*. Moreover, we find that DLG intensity levels at *hiw*+GluRIIA-OE NMJs are significantly increased, similar to the increase observed following GluRIIA-OE at wild-type NMJs. These results are now shown in a revised **Supplemental Figure S6**.

Further, as requested by the Reviewer, we have overexpressed GluRIIB in *hiw*-mutant muscle. GluRIIB-OE at wild-type NMJs essentially phenocopies *GluRIIA* mutants, as GluRIIB-type receptors outcompete GluRIIA-type receptors, as reported previously (Marrus et al., 2004).

[redacted]

9. Reviewer 2 raises a question about the level of sensitivity that mEPSP amplitudes can be measured when they are lowered due to Wnd signaling, and then diminished even further following PhTx application. The Reviewer suggests discussing this issue and/or addressing this concern experimentally through vGlut overexpression.

The Reviewer raises valid points and this is a topic worth addressing. As discussed above, we have attempted to overexpress vGlut in *hiw* mutants, but found vGlut-OE to be incapable of inducing excess vesicular glutamate release with active Wnd signaling (see Point 5 above).

However, we are confident that our quantification of mEPSP amplitude is of sufficient sensitivity to accurately determine the amplitude and to properly evaluate PHP expression, which is the Reviewer's underlying concern. First, the basal noise level in our electrophysiological recordings is +/- 0.03 mV, while the average mEPSP amplitude measured in *hiw*+PhTx is 0.45 mV, well within the range to clearly separate actual mEPSP events from noise. Second, if we were "losing" small mEPSP events in the noise during conditions with small mEPSP events, such as in *hiw*+*GluRIIA* compared to *hiw* alone, then one would expect our average mEPSP frequency to be reduced by the number of events we are missing. However, mEPSP frequency is not statistically significantly different in these conditions (1.47 Hz for *hiw*; 1.64 Hz for *hiw*+*GluRIIA*; see Supplementary Table S1), indicating we are not losing a major number of mEPSP events. Finally, and perhaps most importantly, if PHP were fully expressed or even partially expressed in *hiw*+PhTx or *hiw*+*GluRIIA* compared to *hiw* mutants alone, EPSP amplitude should be the same across all three conditions, in effect stabilizing synaptic strength (that is the definition of PHP). However, as is clearly shown in Figures 4-7, EPSP amplitudes are reduced, proportional to the measured reduction in mEPSP amplitude in each condition. Thus, while we agree with the Reviewer that it is important to be certain about measuring mEPSP amplitudes when studying PHP, we feel there are multiple and independent lines of strong evidence to demonstrate that PHP is not expressed in *hiw* mutants. As suggested by the Reviewer, we discuss these points in the revised **Methods** section (**Page 27 Lines 690-691**). We thank the Reviewer for these thoughtful points and for urging us to more explicitly discuss this issue in the revised manuscript.

10. Reviewer 2 requests that we quantify the number and density of GluRs in *hiw*+Tor-OE, and that we further investigate whether presynaptic Wnd levels are impacted in this manipulation.

We agree that the ability of Tor-OE to increase GluR levels in *hiw* is intriguing. As suggested by the Reviewer, we have now quantified the number and density of GluR puncta in this condition. We find that both the number and density of GluRIIA, B, and D receptor puncta are increased at *hiw*+Tor-OE synapses compared to *hiw*-mutant NMJs alone. This suggests that Tor-OE is capable of increasing expression of the diminished GluRs, perhaps counteracting reduced overall translation in *hiw*-mutant muscle. This data is now shown in **Table S1** of the revised manuscript.

To address the Reviewer's concern about the *hiw*+Tor-OE manipulation potentially impacting presynaptic Wnd signaling in *hiw* mutants, first show that presynaptic growth defects is unchanged in *hiw* compared to *hiw*+Tor-OE (Supplementary Figures S5 and S6). Further, in the revised manuscript, we have utilized an established and sensitive assay reporting Wnd signaling. In particular, nuclear levels of the JNK phosphatase *puckered* (visualized by puc-lacZ staining) have been shown to be dramatically elevated following injury and following activation of Wnd signaling (Xiong et al., 2010). We used this assay as a sensitive measure of Wnd signaling in wild type, Tor-OE, *hiw*, and *hiw*+Tor-OE. We observed a dramatic enhancement of

puc-lacZ staining in *hiw* mutants compared to wild type and Tor-OE. Importantly, puc-lacZ remained enhanced in *hiw*+Tor-OE, demonstrating that postsynaptic Tor overexpression does not measurably influence presynaptic Wnd signaling. This data is now shown in a new **Supplemental Figure S5** and discussed on **Page 13 Lines 403-405** of the revised manuscript.

Minor points:

11. Reviewer 2 requests including data from Figure S2 (panels A and B) in Table S1.

This data was included in Table S1 in the original submission, and we have now added mEPSP frequency and other values to the Table from this data set.

12. Reviewer 2 wonders what effect TNT expression in motor neurons has on postsynaptic DLG and GluR levels in an otherwise wild-type synapse?

This is a very interesting question and certainly is worthy of further investigation.

[redacted]

We have modified the figure detailing these results, which is now **Figure 3**, with it focused on the specific question of whether evoked activity is necessary for the presynaptic overgrowth and postsynaptic GluR levels induced by neuronal Wnd signaling, being careful to normalize all data to an otherwise wild-type synapse expressing TNT.

13. Reviewer 2 questions whether GluRIID levels in Figure 5D may be mislabeled because it is less than GluRIIA and GluRIIB levels?

The Reviewer was correct, and we have now made the changes to correct the mislabeled numbers in **Figure 6** and Table S1. We thank the Reviewer for catching this error.

14. Reviewer 2 requests that we double check the data and labeling in Figure 6G, and to ensure that all normalized and non-normalized data is presented in Table S1.

We apologize for the statistics being mislabeled in the original submission and thank the Reviewer for catching this error. As the Reviewer notes, the mean of *hiw*+GluRIIA-OE+PhTx is not significantly different from baseline (*hiw*+GluRIIA-OE alone), and we have now corrected this error in the revised **Figure 7G**. Further, we have added additional information to detail how statistical tests were performed in the **Methods** section, and have ensured that absolute values of all data are provided in **Table S1** of the revised manuscript.

15. Line 60: Spell out “PHR”.

We have made this change and thank the Reviewer for pointing this out.

16. Line 137: Add “directly” to the indicated sentence.

We have made this change in the revised manuscript and agree that this addition better clarifies the sentence.

17. Line 175: Clarify that AP-evoked “synaptic activity” is meant.

This change has been made.

18. Line 180: Reviewer 2 requests clarification of what is meant by “postsynaptic calcium response”.

We have now included the term “action potential induced neurotransmission”.

[redacted]

19. Figure 8H right panel: Reviewer 2 suggests modifying the schematic as it is confusing in its current form.

We absolutely agree with the Reviewer that the previous schematic was confusing. We have now improved the schematic to more clearly make our point. In the revised **Figure 9H**, we have provided four schematics that illustrate synaptic strength, PHP signaling, and receptor scaling signaling in wild type, wild type+homeostatic plasticity, presynaptic Wnd signaling alone, and presynaptic Wnd signaling stabilized by GluR scaling.

20. Line 544: which experiments are meant by “otherwise specified”?

We apologize for this inaccurate statement. In fact, all electrophysiological recordings in the manuscript were performed in the same 0.3 mM extracellular calcium condition. We have now removed this statement and made this point clear.

Response to Reviewer 3

1. Reviewer 3 raises important mechanistic questions about how DLK signaling impacts the postsynaptic cell and how presynaptic neurotransmitter release is set by this novel pathway?

We agree with the Reviewer that these mechanistic questions are important and deserve further study. In particular, the two questions raised by the Reviewer are of fundamental importance and are of major interest to our lab and others. To address the first question – how presynaptic DLK signaling is communicated to the postsynaptic cell – we have proposed a possible mechanism in the revised Discussion (see **Point 4** response to Reviewer 1 and **Page 19 Lines 560-571**).

[redacted]

The Reviewer also questions how presynaptic function is impacted by DLK signaling and by *hiw*-dependent but DLK-independent signaling. The current manuscript is focused on how the postsynaptic muscle adapts to presynaptic DLK signaling.

[redacted]

2. Reviewer 3 raises the important question about how constitutive DLK signaling may impact synaptic development, and suggests testing PHP and neurotransmission following acute injury or the acute induction of DLK expression later in development.

These are very important and astute points raised by this Reviewer. First, several studies have demonstrated that PHP signaling and expression is not necessarily impacted by disruptions to synaptic function (Frank et al., 2006; Frank et al., 2009) nor in synapse morphology (Dickman and Davis, 2009; Goold and Davis, 2007). Therefore, these studies have established that changes in synapse structure *per se* do not necessarily interfere with PHP expression.

Second, the Reviewer suggests defining synaptic function and PHP during “real” injury. In terms of NMJ function during injury, important work on this question in *Drosophila* was published by Cathy Collins (Mishra et al., 2013; Xiong et al., 2010). Here, her lab developed a “nerve crush” assay and imaged synapse morphology and also determined electrophysiological responses at varying time points. They demonstrated all synaptic transmission at the NMJ ceases within 3-5 hours after nerve crush due to degeneration of the motor neuron. They did observe reduced miniature amplitude, EPSP amplitude, and quantal content following ~3 hours after nerve crush, which is similar to what we demonstrate with chronic DLK signaling in the current manuscript. As suggested by the Reviewer, we have attempted to repeat these experiments, but found that variability of the severity of nerve crush, rate of degeneration, and differences in the overall health of the larvae after injury prevented an analysis that could be properly controlled to examine and interpret PHP signaling in this preparation.

[redacted]

Because we cannot perform the PhTx experiment on the same preparation after recording baseline levels of synaptic transmission in the preparation (Frank et al., 2006), we sought an alternative to further optimization of the nerve crush experiment, given the limited time we had submit the revised manuscript.

However, we did have great success with the second experiment suggested by the Reviewer. Here, the Reviewer requested that we attempt to acutely activate *Wnd* expression at later stages of development. This was a very good idea. We have used the “GeneSwitch” approach, which enables temporal control of gene expression through the use of a modified Gal4 transcription factor that requires the drug RU-486, which can be fed to larvae to initiate transcriptional expression (Osterwalder et al., 2001). There have been reports of “leakiness” of neuronal Geneswitch drivers, so we first tested 4 independent neuronal Geneswitch drivers (BL40981, BL56755, BL56756, and BL43642) with UAS-Tetanus Toxin and assayed viability in the absence of RU-486. We found that one *elav*-GeneSwitch stock (BL40981) had minimal or

no detectable leakiness and went on to attempt the experiment suggested by the Reviewer, feeding third-instar larvae (*elav-GeneSwitch; UAS-wnd*) RU-486 for 24 hrs, 48 hrs, and 72 hrs (see entirely new **Figure 2** now included in the revised manuscript and additional details of the experimental protocol in the revised **Methods** section).

We then assayed synaptic growth, structure, and electrophysiology at each of these time points. Control larvae (same genotype raised in the same conditions in the absence of RU-486) showed no significant difference in synaptic growth or function compared to wild type (Supplementary Table 1 and Figure 2). Further, synaptic growth and structure exhibited no significant difference compared to controls after feeding RU-486 for 24 and 48 hours, while there were some differences after 72 hrs. Finally, mEPSP and EPSP amplitudes were significantly reduced after 24 hrs, and maximally reduced after 48 hrs, a time when synaptic growth and structure was unperturbed. Importantly, PhTx application to larvae fed on RU-486 for 48 hrs resulted in a failure to potentiate presynaptic release (see new **Supplemental Figure 4**). Together, this demonstrates: (1) It is possible to uncouple the extreme changes in synaptic growth and structure from the remodeling of GluRs and inhibition of PHP signaling in the muscle following presynaptic Wnd signaling; (2) the first adaptations to occur following acute activation of presynaptic Wnd signaling are diminishment in postsynaptic GluR levels and inhibition of PHP signaling in the muscle. This is a very important and instructive series of experiments, and we thank the Reviewer for his/her suggestions.

REFERENCES

- Biederer, T., Kaeser, P.S., and Blanpied, T.A. (2017). Transcellular Nanoalignment of Synaptic Function. *Neuron* 96, 680-696.
- Chen, X., and Dickman, D. (2017). Development of a tissue-specific ribosome profiling approach in *Drosophila* enables genome-wide evaluation of translational adaptations. *PLoS Genet* 13, e1007117.
- Daniels, R.W., Collins, C.A., Gelfand, M.V., Dant, J., Brooks, E.S., Krantz, D.E., and DiAntonio, A. (2004). Increased expression of the *Drosophila* vesicular glutamate transporter leads to excess glutamate release and a compensatory decrease in quantal content. *J Neurosci* 24, 10466-10474.
- Davis, G.W., and Goodman, C.S. (1998). Synapse-specific control of synaptic efficacy at the terminals of a single neuron. *Nature* 392, 82-86.
- Dickman, D.K., and Davis, G.W. (2009). The schizophrenia susceptibility gene dysbindin controls synaptic homeostasis. *Science* 326, 1127-1130.
- Frank, C.A., Kennedy, M.J., Goold, C.P., Marek, K.W., and Davis, G.W. (2006). Mechanisms underlying the rapid induction and sustained expression of synaptic homeostasis. *Neuron* 52, 663-677.
- Frank, C.A., Pielage, J., and Davis, G.W. (2009). A presynaptic homeostatic signaling system composed of the Eph receptor, ephexin, Cdc42, and CaV2.1 calcium channels. *Neuron* 61, 556-569.
- Goel, P., Li, X., and Dickman, D. (2017). Disparate Postsynaptic Induction Mechanisms Ultimately Converge to Drive the Retrograde Enhancement of Presynaptic Efficacy. *Cell Rep* 21, 2339-2347.
- Goold, C.P., and Davis, G.W. (2007). The BMP ligand Gbb gates the expression of synaptic homeostasis independent of synaptic growth control. *Neuron* 56, 109-123.
- Li, J., Zhang, Y.V., Asghari Adib, E., Stanchev, D.T., Xiong, X., Klinedinst, S., Soppina, P., Jahn, T.R., Hume, R.I., Rasse, T.M., *et al.* (2017). Restraint of presynaptic protein levels by Wnd/DLK signaling mediates synaptic defects associated with the kinesin-3 motor Unc-104. *eLife* 6.

Marrus, S.B., Portman, S.L., Allen, M.J., Moffat, K.G., and DiAntonio, A. (2004). Differential localization of glutamate receptor subunits at the *Drosophila* neuromuscular junction. *J Neurosci* 24, 1406-1415.

Mishra, B., Carson, R., Hume, R.I., and Collins, C.A. (2013). Sodium and potassium currents influence Wallerian degeneration of injured *Drosophila* axons. *The Journal of neuroscience : the official journal of the Society for Neuroscience* 33, 18728-18739.

Osterwalder, T., Yoon, S.K., White, B.H., and Keshishian, H. (2001). A conditional tissue-specific transgene expression system using inducible GAL4. *PNAS* 98, 12596-12601.

Xiong, X., Wang, X., Ewanek, R., Bhat, P., DiAntonio, A., and Collins, C.A. (2010). Protein turnover of the Wallenda/DLK kinase regulates a retrograde response to axonal injury. *J Cell Biol* 191, 211-223.

REVIEWERS' COMMENTS:

Reviewer #1 (Remarks to the Author):

The authors have addressed my concerns with additional experiments and discussion.

Reviewer #2 (Remarks to the Author):

The authors have addressed all of my concerns satisfactorily and they have done an excellent job revising and further improving the manuscript which will be of interest to many researchers in the community. I recommend its publication in Nature Communications.

Reviewer #3 (Remarks to the Author):

The authors have very adequately addressed my questions. In particular, I was happy to see new figure 2 (geneswitch) to be included in the manuscript and the great results obtained in this experiment.